# Anti-BDCA2 monoclonal antibody inhibits plasmacytoid dendritic cell activation through Fc-dependent and Fc-independent mechanisms

Alex Pellerin[1], Karel Otero[1], Julie M Czerkowicz[1], Hannah M Kerns[1], Renée I Shapiro[2], Ann M Ranger[1], Kevin L Otipoby[1], Frederick R Taylor[2], Thomas O Cameron[2], Joanne L Viney[1] & Dania Rabah[1,*]

## Abstract

Type I interferons (IFN-I) are implicated in the pathogenesis of systemic lupus erythematosus (SLE). In SLE, immune complexes bind to the CD32a (FcγRIIa) receptor on the surface of plasmacytoid dendritic cells (pDCs) and stimulate the secretion of IFN-I from pDCs. BDCA2 is a pDC-specific receptor that, when engaged, inhibits the production of IFN-I in human pDCs. BDCA2 engagement, therefore, represents an attractive therapeutic target for inhibiting pDC-derived IFN-I and may be an effective therapy for the treatment of SLE. In this study, we show that 24F4A, a humanized monoclonal antibody (mAb) against BDCA2, engages BDCA2 and leads to its internalization and the consequent inhibition of TLR-induced IFN-I by pDCs *in vitro* using blood from both healthy and SLE donors. These effects were confirmed *in vivo* using a single injection of 24F4A in cynomolgus monkeys. 24F4A also inhibited pDC activation by SLE-associated immune complexes (IC). In addition to the inhibitory effect of 24F4A through engagement of BDCA2, the Fc region of 24F4A was critical for potent inhibition of IC-induced IFN-I production through internalization of CD32a. This study highlights the novel therapeutic potential of an effector-competent anti-BDCA2 mAb that demonstrates a dual mechanism to dampen pDC responses for enhanced clinical efficacy in SLE.

**Keywords** BDCA2; Fc receptors; immune complexes; plasmacytoid dendritic cells; systemic lupus erythematosus

**Subject Categories** Immunology; Pharmacology & Drug Discovery

## Introduction

Systemic lupus erythematosus (SLE) is a chronic autoimmune inflammatory disease that can affect multiple organs. SLE is characterized by the presence of pathogenic autoantibodies against self-nucleoproteins and DNA (Tan, 1989). Although SLE is a multifactorial disease, increasing evidence indicates that type I interferon (IFN-I) plays a pivotal role in the disease pathogenesis (Crow, 2010); (Elkon & Wiedeman, 2012). IFN-I constitutes a family of cytokines (IFN-α, IFN-β IFN-ε, IFN-κ, IFN-ω, IFN-δ and IFN-τ) that are important for clearance of viral infections but whose uncontrolled production contributes to autoimmune and inflammatory conditions (Theofilopoulos *et al*, 2005). While most cells can produce IFN-I in response to nucleic acids, plasmacytoid dendritic cells (pDCs) are considered the major producers of IFN-I (Liu, 2005).

PDCs are a specialized population of bone marrow-derived cells that can produce as much as a 1,000-fold more IFN-I than other cells in response to ligands that engage the endosomal Toll-like receptor (TLR)7 and TLR9 (Siegal *et al*, 1999). PDCs accumulate in skin lesions of SLE patients as well as in other target organs and have been suggested to be a major source of IFN-I in SLE (Blomberg *et al*, 2001; Farkas *et al*, 2001; Tucci *et al*, 2008; Tomasini *et al*, 2010; Ghoreishi *et al*, 2012). Direct evidence of the critical role of pDCs in SLE pathogenesis has been recently obtained from animal models of SLE in which pDCs were depleted or inactivated (Rowland *et al*, 2014).

Autoreactive immune complexes (IC) are potent inducers of IFN-I in SLE (Lovgren *et al*, 2006). IC bind to low-affinity FcγRIIa (CD32a) Fc receptors on the surface of pDCs and are internalized. In the endosome, single-stranded RNA or DNA containing unmethylated CpG sequences present in these complexes stimulates the nucleic acid sensors TLR7 and TLR9, respectively, leading to the production of IFN-I (Means *et al*, 2005; Lovgren *et al*, 2006). Persistent IC-mediated stimulation of TLR7 and TLR9 in pDCs is suspected to be one of the primary mechanisms whereby pDCs release IFN-I and contribute to SLE disease progression (Ronnblom & Alm, 2001; Swiecki & Colonna, 2010; Ganguly *et al*, 2013).

Blood dendritic cell antigen 2 (BDCA2) is a C-type lectin exclusively expressed on the surface of human pDCs (Dzionek *et al*, 2000). BDCA2 consists of a single extracellular carbohydrate

1   Immunology Research, Biogen Idec, Cambridge, MA, USA
2   Biologics Drug Discovery, Biogen Idec, Cambridge, MA, USA
    *Corresponding author. Tel: +617 679 6255; Fax: +617 679 3208; E-mail: dania.rabah@biogenidec.com

recognition domain, a transmembrane region, and a short cytoplasmic tail that does not harbor any signaling motifs. BDCA2 transmits intracellular signals through an associated transmembrane adaptor, the FcεRIγ, and induces a B-cell receptor (BCR)-like signaling cascade. Antibody-mediated ligation of BDCA2 leads to recruitment of SYK to the phosphorylated immunoreceptor tyrosine-based activation motif (ITAM) of FcεRIγ. SYK activation leads to the activation of BTK and PLCγ2 leading to calcium mobilization. BDCA2 receptor engagement has been shown to inhibit TLR7- or TLR9-induced production of IFN-I and other pDC-derived pro-inflammatory mediators (Dzionek *et al*, 2001; Fanning *et al*, 2006; Cao *et al*, 2007; Rock *et al*, 2007). In addition to inhibiting IFN-I production by pDCs, ligation of BDCA2 with an antibody leads to rapid internalization of BDCA2 by clathrin-mediated endocytosis (Dzionek *et al*, 2000, 2001; Jaehn *et al*, 2008).

In this study, we describe a novel antibody, 24F4A, which binds BDCA2 and leads to its internalization and the consequent inhibition of TLR9-induced IFN-I by pDCs both *in vitro* and *in vivo*. In addition, we describe two distinct mechanisms by which 24F4A can inhibit SLE-IC-mediated pDC activation. The first mechanism is mediated by the induction of signaling from BDCA2, while the second mechanism is mediated by the depletion of Fc receptor (FcR) from the surface of pDCs.

# Results

## Novel anti-BDCA2 monoclonal antibody (mAb) inhibits IFN-I production by pDCs in healthy human donors and SLE patients

BDCA2 ligation with a monoclonal antibody against BDCA2 (clone AC144) has been shown to suppress the ability of human pDCs to produce IFN-I in response to TLR7 and TLR9 ligands (Dzionek *et al*, 2001; Blomberg *et al*, 2003; Cao *et al*, 2007). We generated mouse monoclonal antibodies (mAbs) against human BDCA2 and investigated their ability to inhibit TLR9-induced IFNα by pDCs. Of the 92 hybridomas isolated, only 10 anti-BDCA2 mAbs, belonging to 4 primary-sequence defined families, demonstrated binding to both human and cynomolgus BDCA2 and inhibited TLR9-induced IFNα from human PBMC (Supplementary Table S1 and Supplementary Fig S1). 24F4A demonstrated high potency, comparable to AC144, and was therefore chosen for humanization (Supplementary Table S1 and Supplementary Fig S1). Next, we investigated the potency of 24F4A in whole-blood assays. To this end, whole blood from healthy human donors (*n* = 12) was stimulated with the TLR9 ligand CpG-A in the presence of increasing concentrations of 24F4A. BDCA2 ligation by 24F4A led to a dose-dependent inhibition of TLR9-induced IFNα production (Fig 1A) with an average $IC_{50}$ of 0.06 μg/ml (Fig 1B). 24F4A displayed similar potency in purified peripheral blood mononuclear cells (PBMC) isolated from healthy human donors and SLE patients (Fig 1B). To ensure that 24F4A specifically inhibited pDC-derived IFN-I, PBMC from healthy donors were stimulated with the TLR3 ligand poly(I:C). PDCs do not express TLR3 and therefore do not contribute to IFN-I production induced by poly(I:C) in PBMC culture (Kadowaki *et al*, 2001; Matsumoto *et al*, 2003). As expected, 24F4A inhibited TLR9-induced IFNα but did not impact TLR3-induced IFNα production by PBMC (Supplementary Fig S2).

## Anti-BDCA2-induced BDCA2 internalization is correlated with the inhibition of IFNα production in human pDCs

Antibody-mediated BDCA2 ligation was previously reported to induce rapid receptor internalization (Dzionek *et al*, 2000; Jahn *et al*, 2010). To determine whether 24F4A could induce BDCA2 internalization, whole blood from healthy human donors was incubated overnight with increasing concentrations of 24F4A. Surface expression of BDCA2 was detected on pDCs by flow cytometry using an antibody directed to a non-overlapping epitope of BDCA2 (antibody clone 2D6; Supplementary Fig S3). Treatment with 24F4A led to a dose-dependent decrease in BDCA2 surface expression on pDCs with an average $EC_{50}$ of 0.017 μg/ml (Fig 1C). The $EC_{50}$ of 24F4A-mediated BDCA2 internalization correlated with the $IC_{50}$ of IFNα inhibition (*n* = 10 healthy donors) with an $R^2$ value of 0.68 (Fig 1D). The correlation between BDCA2 internalization and IFN-I inhibition was further supported by experiments using another anti-BDCA2 mAb from our panel, murine 6G6. While 6G6 bound BDCA2 with high affinity (Supplementary Table S1) and achieved full receptor occupancy (Supplementary Fig S4C) it only led to modest BDCA2 internalization and similarly, modest inhibition of TLR9-induced IFNα in PBMC culture (Supplementary Fig S4A and B). Together, these data suggest that BDCA2 internalization and inhibition of TLR9-induced IFN-I production may be mechanistically linked.

## Anti-BDCA2 mAb induces rapid and persistent intracellular localization of BDCA2 and sustained inhibition of IFN-I

Studies of 24F4A-mediated BDCA2 internalization kinetics showed that BDCA2 surface expression rapidly decreased following treatment with 1 μg/ml 24F4A, reaching background levels within 1 h of treatment. Treatment with tenfold lower concentration of 24F4A delayed surface BDCA2 downmodulation by 2 h, while a 100-fold lower concentration of 24F4A was able to cause significant receptor internalization only after overnight culture (Fig 2A). These data indicate that BDCA2 is rapidly internalized upon ligation with 24F4A with dose-dependent kinetics.

Next, to visualize the internalization of BDCA2 and subcellular localization after ligation with 24F4A, purified pDCs were incubated with AlexaFluor647-labeled 24F4A and analyzed by confocal microscopy. As shown in Fig 2B, labeled 24F4A bound to BDCA2 was localized on the cell surface of pDCs at 4°C. After a 10 min incubation at 37°C, labeled 24F4A was detected inside pDCs presumably complexed with BDCA2. After a 1 h incubation at 37°C the 24F4A/BDCA2 complex was localized in the LAMP1[+] endolysosomal compartments where it persisted for up to 14 h post-treatment (Fig 2B).

While it has been shown that anti-BDCA2 mAb can lead to BDCA2 internalization and accumulation in intracellular compartments (Jaehn *et al*, 2008), it is unknown whether BDCA2 is recycled to the cell surface after 24F4A-mediated internalization. To address this question, whole blood was pre-incubated in the absence or presence of 10 μg/ml of 24F4A for 1 h at 37°C to obtain maximal internalization. PBMC were subsequently isolated, thereby removing unbound 24F4A that could engage re-expressed BDCA2. After 16 h at 37°C, BDCA2 levels remained low on pDCs that were pre-incubated with 24F4A but cultured in the absence of 24F4A and were comparable to the BDCA2 levels on pDCs in control cultures

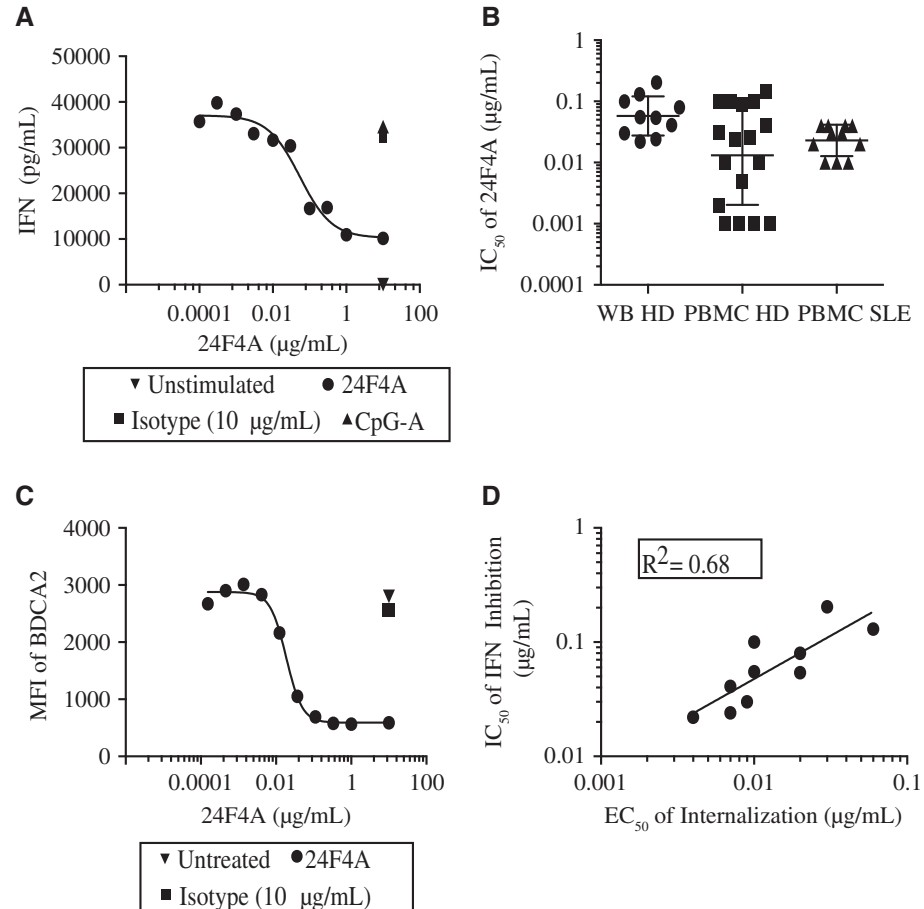

**Figure 1.  Anti-BDCA2 mAb inhibits IFNα production by pDCs and induces internalization of BDCA2 on the surface of pDCs.**

A, B   Whole blood or PBMC were treated with increasing concentrations of 24F4A and stimulated with CpG-A for 16 h at 37°C. Anti-BDCA2-mediated IFNα inhibition in whole-blood assays (A). IFNα levels were detected using human IFNα ELISA. Graph depicts average of duplicate wells of one representative donor ($n = 10$). $IC_{50}$ of 24F4A-mediated IFNα inhibition (B) in whole blood (circles) ($n = 10$) and PBMC ($n = 18$) from healthy human donors (squares) or SLE patients (triangles) ($n = 11$). Horizontal bar represents the mean $IC_{50}$ for each sample type. Error bars represent SD of $IC_{50}$ between donors.

C   Anti-BDCA2-mediated internalization in whole-blood assays. Whole blood was treated with increasing concentrations of 24F4A for 16 h. Mean fluorescence intensity (MFI) of BDCA2 was determined with a non-cross-blocking anti-BDCA2 mAb (2D6). Shown is a representative plot of 10 experiments conducted.

D   The $IC_{50}$ of 24F4A-mediated IFNα inhibition was compared to the $EC_{50}$ of 24F4A-induced BDCA2 internalization.

that were continuously exposed to 10 μg/ml of 24F4A (Fig 2C). Furthermore, when stimulated with CpG-A, IFNα production was inhibited in cells pre-incubated with 24F4A to levels indistinguishable from that seen in PBMC continuously exposed to 24F4A (Fig 2D). These data demonstrate that BDCA2 is not rapidly recycled to the cell surface of pDCs after 24F4A-mediated BDCA2 internalization. Furthermore, the data show that 1 h pre-incubation with 24F4A is sufficient to inhibit TLR9-induced IFNα production. We extended these studies to ascertain whether internalized BDCA2 was still capable of inhibiting IFNα over longer pre-incubation periods with 24F4A. To this end, whole blood from healthy human donors was pre-incubated with 24F4A at 37°C for various periods up to 9 h, followed by stimulation with CpG-A. As shown in Fig 2E, pre-incubation with 24F4A for up to 9 h led to maximal inhibition of TLR9-induced IFNα production. Our data suggest that the persistent localization of the 24F4A/BDCA2 complex in LAMP1[+] endolysosomal compartment in the absence of recycling could be important for its ability to mediate IFN-I inhibition.

## Anti-BDCA2 mAb leads to BDCA2 internalization and IFN-I inhibition *in vivo* in cynomolgus monkeys

We next tested the pharmacokinetic properties and biological activity of 24F4A *in vivo*. Since rodents do not express BDCA2 (Dzionek *et al*, 2000), we performed these studies in the cynomolgus monkey. Cynomolgus monkey BDCA2 protein shares 90.6% homology with human BDCA2. 24F4A binds similarly to cynomolgus and human BDCA2 with an average $EC_{50}$ of 0.63 μg/ml ($n = 5$) and 0.7 μg/ml ($n = 8$), respectively (Supplementary Fig S5).

Nine cynomolgus monkeys were divided into three groups that received a single intravenous (IV) injection of vehicle (sodium citrate buffer), 1 mg/kg 24F4A, or 10 mg/kg 24F4A. Animals were bled at various time points before and after 24F4A administration. First, we addressed whether administration of 24F4A leads to BDCA2 internalization *in vivo*, using flow cytometry. Because the 2D6 anti-BDCA2 clone does not cross-react with cynomolgus BDCA2 (Supplementary Table S1), a two-step approach was used

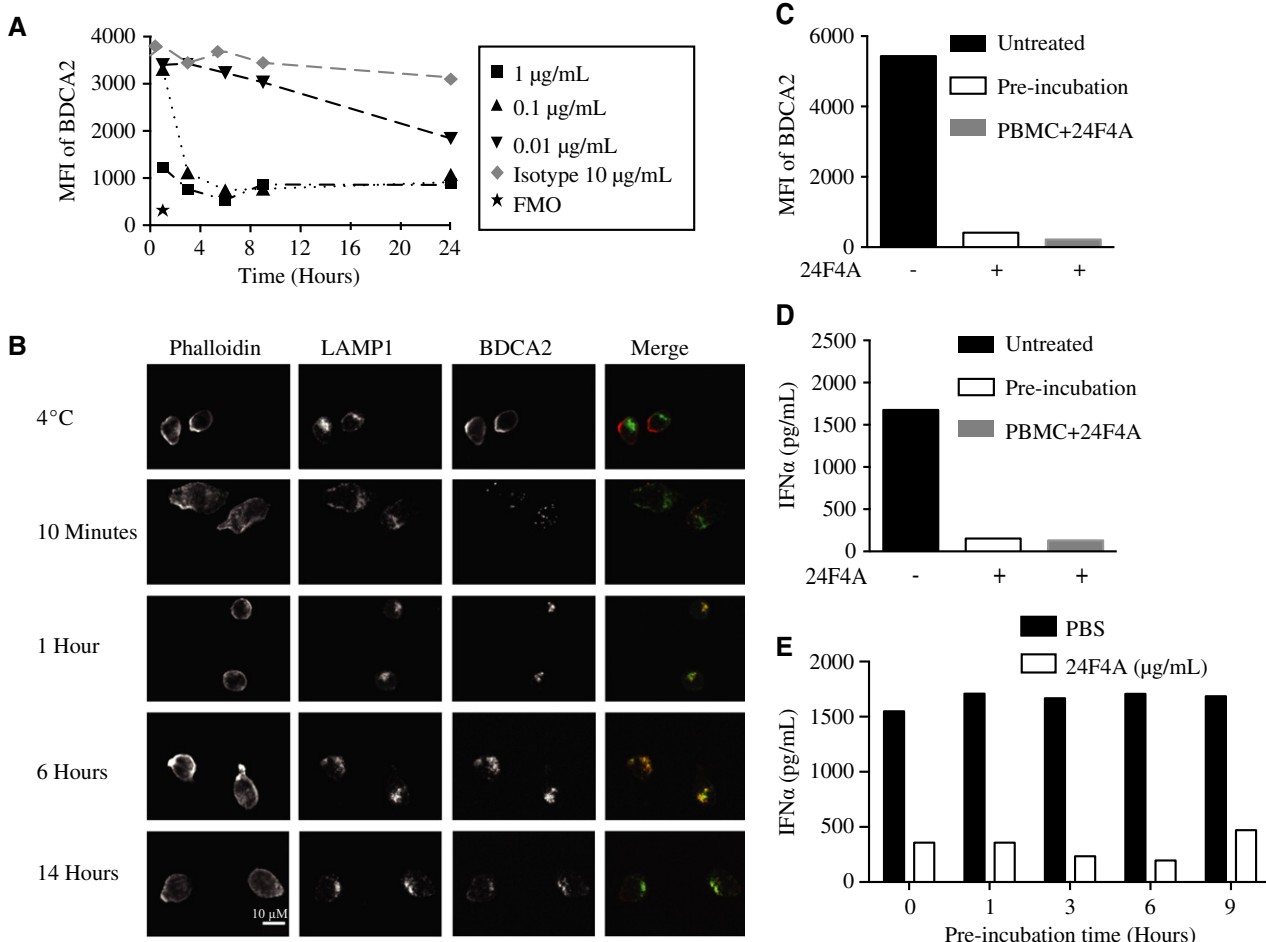

**Figure 2. 24F4A induces rapid internalization of BDCA2 resulting in trafficking to LAMP1$^+$ compartments and sustained IFNα inhibition.**

A Whole blood was treated with the indicated concentrations of 24F4A or the IgG1 isotype control antibody for 0, 1, 3, 6, 9, and 24 h. FACS analysis was performed to determine BDCA2 levels on pDCs. Fluorescence minus one (FMO) represents background staining of BDCA2. Shown is a representative plot of three independent experiments.

B Isolated pDCs were treated with 10 μg/ml 24F4A-AF647 at 4°C or at 37°C and analyzed at the indicated time points. Subcellular localization of BDCA2 (red) compared to LAMP1 (green) was assessed using confocal microscopy. Phalloidin was used to delineate the cell membrane. Yellow represents co-localization of BDCA2 and LAMP1. Shown is a representative image of four experiments conducted.

C, D Whole blood was either treated with 24F4A for 1 h at 37°C (pre-incubation) or left untreated. PBMC were isolated from each condition. PBMC from the untreated whole blood were subsequently treated with 10 μg/ml 24F4A, and cells from all conditions were stimulated with CpG-A for 16 h. Plot represents mean of duplicate wells. Flow cytometry was used to measure BDCA2 levels (MFI). Shown is a representative donor of eight donors tested.

E Whole blood was pre-treated for 0, 1, 3, 6, and 9 h with 10 μg/ml of 24F4 or the isotype control and then stimulated with CpG-A for additional 16 h. IFNα levels were measured by ELISA. Shown is a representative plot of two independent experiments.

Source data are available online for this figure.

to detect internalization of BDCA2 on cynomolgus monkey pDCs. Unoccupied surface BDCA2 was detected on pDCs in whole blood using fluorescently labeled 24F4A (direct method), while surface BDCA2 bound to 24F4A was detected using a fluorescently labeled anti-human IgG1 (indirect method). The lack of unoccupied BDCA2 (direct method) coupled with loss of detectable 24F4A (indirect method) indicated BDCA2 internalization. Results from a representative animal from both the vehicle-treated group and the 1 mg/kg 24F4A-treated group are shown in Fig 3A and B. Prior to vehicle and 24F4A administration, the baseline surface expression of BDCA2 was assessed for each cynomolgus monkey using the direct method (Fig 3A-i and B-i, dotted red line). In addition,

maximal binding of BDCA2 to 24F4A was established prior to 24F4A administration by "spiking" whole blood with saturating amounts of 24F4A *in vitro* and measuring bound 24F4A by the indirect method (Fig 3A-ii and B-ii, solid red line). Within 6 h of 24F4A administration at 1 mg/kg, BDCA2 expression on the surface of pDCs decreased to almost undetectable levels (Fig 3B-iii, dotted red line) but not in the vehicle-treated group (Fig 3A-iii, dotted red line). In addition, the levels of bound 24F4A (Fig 3B-iv, solid black line) were indistinguishable from the vehicle-treated group (Fig 3A-iv, solid black line). The lack of available BDCA2 receptor together with the lack of detectable 24F4A on the surface of pDCs indicated internalization of BDCA2.

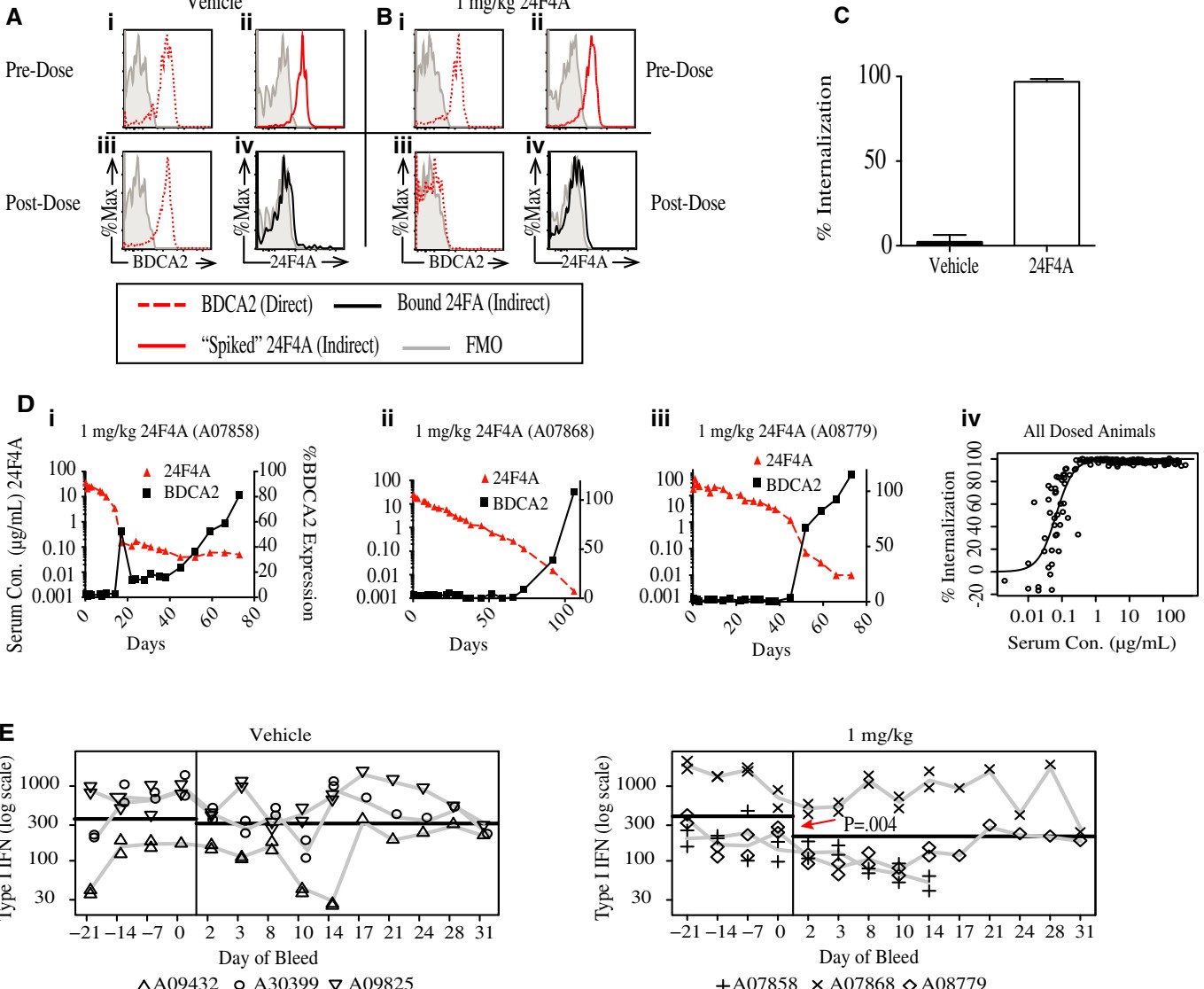

**Figure 3. 24F4A mediates BDCA2 internalization and type I IFN inhibition *in vivo*.**

Cynomolgus monkeys were administered 24F4A (10 or 1 mg/kg) or vehicle (*n* = 3 for each dose group) intravenously. Cynomolgus monkeys were bled at various time points, and flow cytometry was used to measure BDCA2 expression and receptor occupancy. PDCs were defined as CD20⁻, CD14⁻, CD123⁺, and HLADR⁺.

A–C   Prior to *in vivo* dosing, baseline surface levels of BDCA2 for both the vehicle (Ai) and 1 mg/kg (Bi) animals (red, dotted line) were established by staining with fluorescently labeled 24F4A (direct method). Maximal binding of 24F4A to BDCA2 was also established pre-dose in the vehicle (Aii) and 1 mg/kg (Bii) animals (red, solid line) by treating whole blood with 10 µg/ml of 24F4A at 4°C and then detecting bound 24F4A with a fluorescently labeled anti-human IgG1 (indirect method). The direct method was used to stain whole blood from both the vehicle (Aiii) and 1 mg/kg 24F4A (Biii) animals 6 h post-dose (red, dotted line). In a separate stain, the indirect method was used to detect bound 24F4A in the vehicle (Aiv) and 1 mg/kg (Biv) treated animals (black, solid line). (C) Percent BDCA2 internalization relative to pre-dose BDCA2 levels 6 h post-dose with vehicle, 10 mg/kg, or 1 mg/kg 24F4A. Graph shows mean ± standard deviation for each group (*n* = 3).

D   PK/PD relationship between 24F4A serum concentrations (red triangle, left axis) and BDCA2 expression on pDCs (black squares, right axis, normalized to pre-dose levels) from the 1 mg/kg group (i–iii). Serum 24F4A was measured by ELISA. (iv) Percent BDCA2 internalization versus serum concentration of 24F4A for all dosed cynomolgus monkeys at all time points tested.

E   Whole blood from vehicle- or 1 mg/kg 24F4A-treated monkeys was stimulated with CpG-A, and induction of IFN-I was measured by MxA bioassay at various time points pre- and post-treatment. Horizontal black lines represent the model-based estimates of the geometric mean of IFN-I in pre- and post-dose samples. Duplicate symbols represent independent replicates of the MxA bioassay for that time point. Statistical analysis was performed using a two-way mixed-effects analysis of variance (ANOVA).

Over 95% of surface BDCA2 was internalized in all animals within 6 h of IV treatment (1 and 10 mg/kg) (Fig 3C). Internalization of BDCA2 correlated with circulating levels of 24F4A,

establishing a pharmacokinetic/pharmacodynamic (PK/PD) relationship *in vivo*. When 24F4A serum concentrations decreased to a range of 0.1–0.03 µg/ml, the level of BDCA2 recovered to

> 70% of the baseline level (Fig 3D-i–iii), establishing an $EC_{50}$ of 0.133 μg/ml (Fig 3D-iv).

Next, we determined whether 24F4A, when administered to cynomolgus monkeys *in vivo*, could inhibit the production of IFN-I after TLR9 stimulation *ex vivo*. As the reagents for stimulating and detecting cynomolgus IFN-I are not adequate, we utilized an established MxA bioassay (Wadhwa *et al*, 2013) to indirectly measure IFN-I production from CpG-A-stimulated whole blood. A two-way mixed-effects analysis of variance (ANOVA) was used to estimate and compare the mean $log_{10}$ IFN-I concentrations for each treatment group before and after administration of 24F4A up to day 31 or prior to loss of the pharmacodynamics effect (BDCA2 internalization). Vehicle injection did not significantly affect the level of IFN-I, while 1 mg/kg 24F4A led to a 46% decrease in IFN-I production (95% CI: 18–65%; *P* = 0.004) (Fig 3E). IFN-I production was also significantly decreased following 10 mg/kg IV administration (52%, 95% CI: 22–70%; *P* = 0.003) (Supplementary Fig S6). There was no significant change in the frequency of circulating pDCs after 24F4A administration (Supplementary Fig S6). Together, these results demonstrate for the first time that administration of single dose of anti-BDCA2 mAb in cynomolgus monkeys causes rapid internalization of BDCA2 and a significant reduction in TLR9-induced IFN-I production in *ex vivo* whole-blood assays.

### The Fc region of anti-BDCA2 mAb enhances the inhibition of immune complex-induced IFN-I production by human pDCs

Bivalent binding of anti-BDCA2 mAb to BDCA2 is an essential requirement for the agonistic activity of the mAb. Monovalent Fab fragments do not elicit BDCA2 signaling and do not inhibit TLR7 or TLR9-induced IFN-I production by pDCs (Jahn *et al*, 2010). It is not known, however, whether Fc engagement of 24F4A contributes to the cross-linking of BDCA2, thereby enhancing its ability to inhibit IFN-I. To address this possibility, we generated an Fc-effectorless form of 24F4A (24F4A-ef) with a human IgG4.P/human IgG1 chimeric Fc region that cannot bind Fc receptors (Supplementary Fig S7). PDCs were incubated in the presence of increasing concentrations of 24F4A or 24F4A-ef and stimulated with synthetic TLR7 ligand (R848), synthetic TLR9 ligand (CpG-A), or a disease-relevant ligand, SLE immune complexes (SLE-IC), Sm/RNP. Sm/RNP autoantigens, consisting of U1 RNA bound by Smith antigen (Sm), are frequently targeted by autoantibodies in SLE and are capable of activating TLR7 (Lau *et al*, 2005; Vollmer *et al*, 2005; Christensen *et al*, 2006). 24F4A and 24F4A-ef were equipotent at inhibiting R848 and CpG-A-induced IFNα by pDCs (Fig 4A–D). In contrast to the synthetic TLR7 or TLR9 ligands, 24F4A was 40-fold more potent at inhibiting Sm/RNP IC-induced IFNα with an average $IC_{50}$ of 0.03 μg/ml, compared to 24F4A-ef, which inhibited with an average $IC_{50}$ of 1.3 μg/ml (Fig 4E and F). Treatment with effector-competent isotype control antibody at the highest concentration of 10 μg/ml did not affect the induction of IFNα by R848, CpG-A, or SLE-IC (Sm/RNP) (Fig 4A, C and E). The difference in potency of 24F4A versus 24F4A-ef was not due to a difference in signaling downstream of BDCA2 engagement as 24F4A and 24F4A-ef were equally capable of inducing phosphorylation of Syk and PLCγ2 (Fig 4G).

While the uptake of synthetic TLR ligands is independent of Fc receptors (Vollmer *et al*, 2004), IC need to be internalized by

CD32a (FcγRIIa) (Means *et al*, 2005). Although CD32a was reported to be the only Fc receptor expressed on pDCs (Bave *et al*, 2003), recent transcript profiling data indicated that CD32b could be expressed at similar levels in pDCs (Guilliams *et al*, 2014). To address this question, we evaluated the transcript levels of CD32a and CD32b in isolated pDCs using Q-PCR and confirmed the exclusive expression of CD32a in pDCs (Supplementary Fig S8). As expected, CD32a blockade led to complete inhibition of IC-mediated IFNα production by pDCs, but did not affect IFNα production mediated by synthetic TLR7 or TLR9 ligands (CpG-A or R848) (Fig 4H). Taken together, these data show that while the Fc domain of 24F4A did not impact the signaling activity of the mAb and its inhibitory activity on synthetic TLR ligands, it contributed to the potent inhibition of pDC responses to TLR ligands that are dependent on Fc receptors.

### Anti-BDCA2 mAb leads to the concurrent internalization of BDCA2 and CD32a

The impact of the Fc domain of 24F4A on IC-mediated IFN-I production by pDCs suggested the intriguing possibility that 24F4A could modulate the surface expression of CD32a. To investigate this possibility, we isolated pDCs and treated them with increasing concentrations of 24F4A or 24F4A-ef and assessed the surface expression of BDCA2 and CD32a by flow cytometry. 24F4A-ef was as potent as 24F4A at inducing BDCA2 internalization (Fig 5A and B); however, only 24F4A induced downmodulation of CD32a on the surface of pDCs (Fig 5C and D). Similar results were obtained with the effector-competent anti-BDCA2 mAb clone AC144 (Supplementary Fig S9). Treatment with effector-competent isotype control at the highest concentration of 10 μg/ml had no effect on CD32a surface levels (Fig 5C and D). In addition, a humanized anti-CD40 mAb that binds CD40 on the surface of pDCs and harbors an identical Fc region to that of 24F4A did not result in downmodulation of CD32a (Fig 5E). To ensure that the lack of effect of anti-CD40 mAb on CD32a levels was not due to lower CD40 expression compared to CD32a, we used QuantiBRITE beads to measure the density of BDCA2, CD40 and CD32a on the surface of pDCs. Under the same experimental conditions, both CD40 and BDCA2 were expressed at higher level than CD32a on pDCs (Supplementary Fig S10).

Next, we investigated whether the downmodulation of CD32a with an effector-competent mAb was coupled with BDCA2 internalization. We took advantage of the fact that murine 6G6 mAb does not mediate significant BDCA2 internalization (Supplementary Fig S4), and we generated a chimera of 6G6 (Chi-6G6) that has a human Fc region identical to that of 24F4A. We first ensured that Chi-6G6 was still able to bind BDCA2 (Supplementary Fig S4C). We then tested the impact of Chi-6G6 on BDCA2 and CD32a levels on isolated pDCs. Chi-6G6 led to only marginal BDCA2 internalization and similarly led to marginal CD32a downmodulation (Fig 5A–D). 24F4A also mediated BDCA2 internalization and CD32a downmodulation with similar potency (Fig 5F). Collectively, these data suggest that only effector-competent anti-BDCA2 mAbs capable of mediating significant BDCA2 internalization, such as 24F4A, can lead to CD32a internalization.

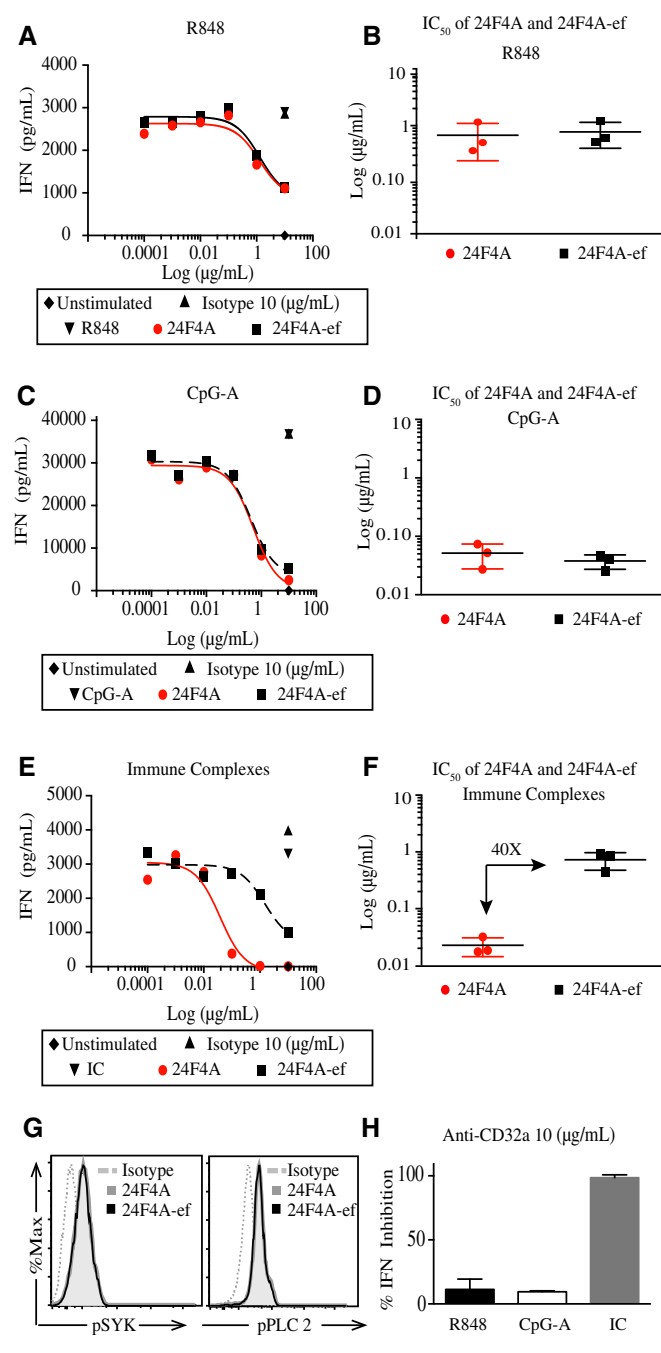

**Figure 4. The Fc region of 24F4A enhances the inhibition of immune complex-induced IFNα production by human pDCs.**

A–F  Isolated human pDCs were treated with increasing concentrations of 24F4A (red circles), 24F4A-ef (black squares), or 10 μg/ml of the isotype control (black triangle) and stimulated with 5 μM of R848 (A and B), 1 μM of CpG-A (C and D), or immune complexes (E and F) for 16 h at 37°C. IFNα concentrations in cultured supernatants were determined using ELISA. Representative plots for each stimulation (A, C and E) and the average $IC_{50}$ of 24F4A and 24F4A-ef for each stimulation condition (B, D and F) are shown (n = 3). Horizontal bars represent mean $IC_{50}$, and error bars represent SD of the mean from three independent experiments.

G  Isolated human pDCs were treated with 10 μg/ml of 24F4A (black line), 24F4A-ef (gray shaded histogram), or the isotype control (gray dotted line) for 10 min at 37°C. Flow cytometry was used to evaluate pSYK and pPLCγ2. Shown is a representative plot of two independent experiments.

H  Isolated pDCs treated with 10 μg/ml of anti-CD32a (AT10) and stimulated with CpG-A, R848, or immune complexes. Error bars represent SD of percent inhibition of IFNα from three independent experiments.

## Discussion

In this study, we report the generation of an anti-BDCA2 mAb, 24F4A, that can potently inhibit TLR7 and TLR9-induced IFN-I in human pDCs from healthy and SLE patients, as has been shown with a previously described anti-BDCA2 mAb (Dzionek et al, 2001; Blomberg et al, 2003; Fanning et al, 2006; Cao et al, 2007; Rock et al, 2007). Importantly, 24F4A specifically inhibited TLR7 and TLR9-induced IFN-I by pDCs and did not impact IFN-I production by other cell types triggered with a different TLR ligand. Several approaches for targeting the IFN-I pathway are currently being tested in clinical trials including neutralizing antibodies against

IFNα or IFN α,β receptor blockade (Lichtman et al, 2012; Kirou & Gkrouzman, 2013). While these strategies could lead to global IFN-I blockade, functional inhibition of pDCs by 24F4A is expected to dampen pDC-derived IFN-I during disease, while preserving the protective IFN-I response to viruses in other immune and stromal cells (Swiecki & Colonna, 2010; Wang et al, 2012) which could be a safer therapeutic approach.

24F4A led to rapid internalization of BDCA2 and its retention in endolysosomal compartments for at least 14 h with no evidence of recycling to the cell surface. Once inside the cell, 24F4A/BDCA2 complex was capable of meditating maximal inhibition of TLR9-induced IFNα production by pDCs for prolonged periods. A direct correlation was observed between the potency of anti-BDCA2-mediated internalization of BDCA2 and the inhibition of TLR9-induced IFNα production by pDCs, further supporting that the internalization and IFNα inhibition are linked. This mechanism of action of 24F4A could be clinically beneficial as it suggests that treatment with 24F4A might functionally inhibit pDCs for as long as the 24FA/BDCA2 complex is localized in intracellular compartments. Future work will address whether co-localization of BDCA2 with TLR7 and TLR9 in endolysosomal compartments is necessary for inhibition of TLR7 or TLR9 signaling and IFN-I production by pDCs.

BDCA2 is uniquely expressed on pDCs in human and non-human primate. We chose the cynomolgus monkey to gain insight into the pharmacokinetic properties and biological activity of 24F4A in vivo. Single-dose systemic administration of 24F4A to cynomolgus monkeys led to rapid and sustained internalization of BDCA2 from the surface of pDCs. Receptor internalization correlated with circulating 24F4A levels, thereby establishing a PK/PD correlation in vivo. Despite inter- and intra-animal variability, 24F4A-treated animals displayed significantly lower levels of IFN-I following ex vivo stimulation of whole blood with CpG-A. Importantly, the reduction in IFN-I post-treatment was not due to changes in pDC numbers, as treatment with 24F4A did not alter the number of circulating pDCs. Even though 24F4A is an effector-competent mAb, the rapid, sustained, and near-complete internalization of BDCA2 after 24F4A administration could explain the lack of antibody-mediated pDC depletion in vivo.

In vivo treatment with anti-BDCA2 mAb, AC144, has been shown to inhibit pDC-derived IFNα production and improve skin

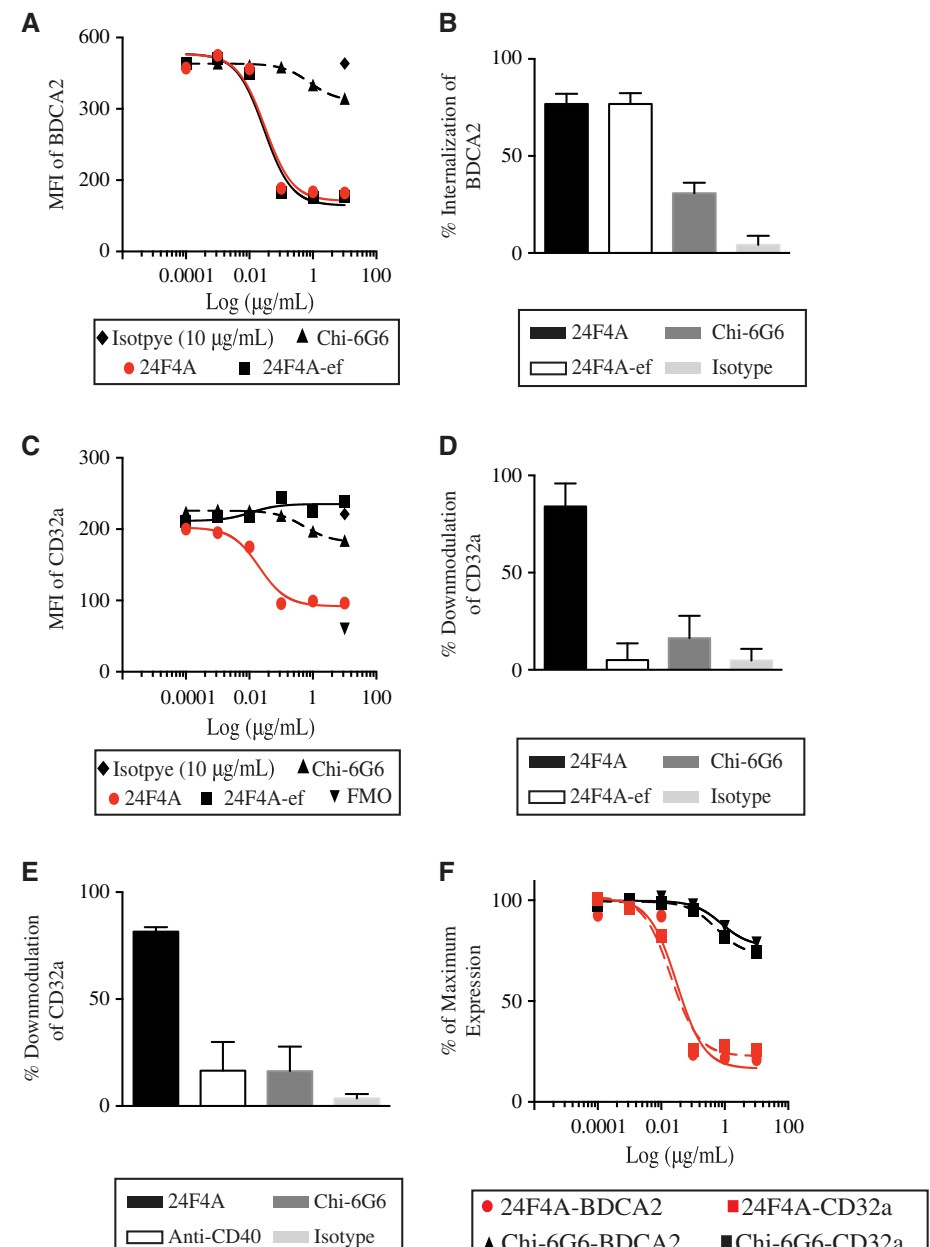

**Figure 5.  Treatment with 24F4A leads to a dose-dependent downmodulation of CD32a on pDCs.**

A–F  Isolated pDCs were treated with increasing concentrations of 24F4A (red circles), 24F4A-ef (squares), Chi-6G6 (triangle), or isotype control (diamond) for 16 h at 37°C and stimulated with CpG-A. Flow cytometry was performed to evaluate the MFI of (A) BDCA2 and (C) CD32a on the surface of pDCs. Percent internalization of BDCA2 (B) and CD32a (D) on pDCs treated with 10 μg/ml of 24F4A, 24F4A-ef, Chi-6G6, or isotype control. Error bars represent SD of percent internalization of five experiments conducted. (E) Percent CD32a downmodulation on isolated pDCs treated with 10 μg/ml of an anti-CD40 mAb. Error bars represent SD of percent downmodulation of two independent experiments. FMO represents background staining of CD32a (inverted triangle). (F) Percent of maximal expression of BDCA2 or CD32a in the presence of increasing concentrations of 24F4A or Chi-6G6. Maximal expression was defined by the MFI of both BDCA2 and CD32a on untreated cells. Shown is a representative plot of five independent experiments conducted.

disease in a human psoriatic xenograft model (Nestle *et al*, 2005). Our data from the cynomolgus monkey confirm the ability of anti-BDCA2 mAbs to functionally inhibit pDCs *in vivo* and provide the first evidence that an effector-competent anti-BDCA2 mAb can induce BDCA2 internalization *in vivo* without leading to pDC depletion.

The fact that 24F4A leads to functional inhibition of pDCs *in vivo* rather than cell depletion could be therapeutically advantageous. Sisirak *et al* demonstrated that even a partial functional inhibition of pDCs can drastically improve lupus-like disease in mouse models of SLE (Sisirak *et al*, 2014). In contrast, complete pDC depletion has been shown to impact anti-viral immunity (Cervantes-Barragan

et al, 2012). Therefore, functional inhibition of pDCs through BDCA2 ligation in the absence of pDC depletion is a unique approach that could lead to both efficacy and a better safety profile in autoimmune diseases such as SLE.

Cross-linking BDCA2 with anti-BDCA2 mAb promotes a BCR-like signaling cascade resulting in the inhibition of TLR7 or TLR9-mediated IFN-I production by pDCs (Dzionek et al, 2001; Fanning et al, 2006; Cao et al, 2007; Rock et al, 2007). Bivalent binding of anti-BDCA2 mAb through the F(ab')$_2$ region is an essential requirement for the agonistic inhibitory activity of the mAb (Jahn et al, 2010). Consistent with these observations, effector function-deficient 24F4A (24F4A-ef) was as potent as 24F4A in mediating downstream signaling and internalization of BDCA2. 24F4A-ef was also equally potent at inhibiting IFN-I production from pDCs stimulated with synthetic ligands for TLR7 or TLR9. While the Fc region of 24F4A does not play a critical role in BDCA2 signaling, we found that it potentiates the inhibition of pDCs in response to SLE-IC. 24F4A was 40-fold more potent in inhibiting IC-induced IFNα by pDCs compared to 24F4A-ef. Mechanistically, 24F4A, but not 24F4A-ef, led to the downmodulation of CD32a on the surface of human pDCs. Since pDC activation by SLE-IC is dependent on CD32a expression (Means et al, 2005), our data show that the Fc region of 24F4A improves the efficacy of 24F4A by depleting CD32a from the cell surface and preventing the stimulation of pDCs by SLE-IC.

It was recently shown that the Fc region of IgG inhibits SLE-IC-induced IFN-I at high concentrations of IgG (> 50 μg/ml). This effect was due to the inhibition of the binding of SLE-ICs to CD32a on pDCs and did not involve downmodulation of CD32a (Wiedeman et al, 2013). Consistent with this, we found that IgG1 isotype had no effect on SLE-IC-mediated IFN-I production by pDCs at concentrations where 24F4A led to complete inhibition of IFN-I production. This suggests that the impact of Fc region of 24F4A on IFN-I production is primarily due to the loss of CD32a from the cell surface rather than blockade of the IC-binding sites on CD32a.

Collectively, these data suggest that 24F4A, not 24F4A-ef, simultaneously engages BDCA2 and CD32a leading to the internalization of both receptors. This observation is consistent with the previously described "scorpion" mechanism wherein an antibody recognizing a cell surface molecule via its F(ab')$_2$ region can simultaneously engage an FcR on the same cell via its Fc region, thus creating a trimolecular complex (Kurlander, 1983; Dunn-Siegrist et al, 2007; Hogarth & Pietersz, 2012; Shang et al, 2014). Our study describes the first example where the functional outcome of the putative "scorpion" mechanism is to co-internalize the FcR (CD32a) with the primary binding target (BDCA2). Only effector-competent anti-BDCA2 mAbs capable of internalizing BDCA2 lead to CD32a downmodulation. A non-internalizing anti-BDCA2 antibody (Chi-6G6) that harbors an identical Fc region to that of 24F4A was not capable of internalizing CD32a. Attempts to study co-localization of intracellular CD32a and BDCA2 in primary human pDCs by confocal microscopy were unsuccessful with available reagents to detect CD32a. More work will need to be done to delineate the exact details of the mechanism leading to concurrent internalization of BDCA2 and CD32a.

In conclusion, we show that BDCA2 engagement with the mAb 24F4A elicits a dual mechanism to dampen pDC responses. In a disease setting such as SLE, the F(ab')$_2$ region of 24F4A should inhibit SLE-IC-induced IFN-I production by pDCs through ligation

and internalization of BDCA2. Simultaneously, the Fc region of 24F4A should lead to CD32a downmodulation, thereby eliminating the receptor for immune complexes on pDCs and preventing the pathological activation of naïve or newly generated pDCs. Current therapeutic approaches are accounting for FcRs as pertinent targets for the treatment of inflammatory diseases (Hogarth & Pietersz, 2012). We propose that effector-competent antibodies to endocytic receptors expressed on immune cells could efficiently target Fc receptors through the same scorpion mechanism as BDCA2 and offer novel therapeutic approaches for inflammatory and autoimmune diseases.

## Materials and Methods

### 24F4 mAb generation

Eight- to 10-week-old BALB/c female mice were purchased from The Jackson Laboratory. Mice were housed and maintained in an AAALAC-accredited facility under standard conditions. All procedures were performed in accordance with guidelines set forth by the Biogen Idec IACUC under an approved protocol. The 24F4 murine hybridoma was derived from a BALB/c mouse immunized using Gene Gun with a mammalian expression vector which co-expresses full-length human BDCA2 and FcεRIγ cDNAs. The mAb was humanized to produce an effector-competent form 24F4A (human wt IgG1) and an effectorless form, 24F4A-ef (human IgG4.P/human IgG1 chimeric). Detailed methods are presented in the Supplementary Materials and Methods.

### Human blood collection and PBMC isolation

All experiments using human blood were done in accordance with IRB protocol 20121572. Human peripheral blood from healthy donors was obtained from Blood Donor Program at Biogen Idec (Cambridge, MA). All experiments conformed to the principles set out in the WMA Declaration of Helsinki and the Department of Health and Human Services Belmont Report. Blood from SLE patients was obtained from Sera Care Life Sciences (Milford, MA). Human venous whole blood was collected in sodium heparin tubes. PBMC were isolated using Ficoll gradients (GE Healthcare). Whole blood was overlaid onto Ficoll and centrifuged for 20 min at 600 g with no brake. Cells were washed in PBS and counted on a Vi-cell (Beckman Coulter).

### Plasmacytoid dendritic cell isolation

Buffy coats from healthy donors were obtained from Research Blood Components (Boston, MA). PBMC were isolated as previously described. PDCs were isolated from PBMC using the Diamond pDC Isolation kit II (Miltenyi) according to the manufacturer's protocol. Briefly, PBMC were labeled with a biotinylated non-pDC antibody cocktail and then with anti-biotin magnetic beads to negatively select PDCs. PDCs were then positively selected using anti-BDCA4 beads. Purity was assessed by staining with labeled anti-CD123 (7G3, BD Biosciences) and anti-BDCA2 (AC144, Miltenyi) and measured by flow cytometry. Purity was routinely > 90%. Cells were acquired on a LSRII or Fortessa and analyzed in FlowJo.

### *In vitro* simulations and IFNα measurement

#### *Whole blood*

Whole blood collected in sodium heparin tubes was plated in 96-well round-bottom plates. The blood was treated with 10, 3.33, 1.11, 0.37, 0.124, 0.04, 0.014, 0.005, 0.0015 and 0.005 µg/ml of 24F4A or 10 µg/ml of an isotype control (human IgG1, Biogen) and simultaneously stimulated with 200 µg/ml of CpG-A (2216, Invivogen) and cultured for 16 h at 37°C and 5% $CO_2$. All conditions were set up in duplicate. Serum was collected, and IFNα was evaluated.

#### *PBMC*

PBMC were plated at $1 \times 10^6$/well in complete RPMI media (10% FBS, 1× non-essential amino acids, 1× Pen-Strep, 1 mM sodium pyruvate, 10 mM HEPES, 50 µM 2-mercaptoethanol and 2 mM L-glutamine) in 96-well round-bottom plates and treated with 10, 3.33, 1.11, 0.37, 0.124, 0.04, 0.014, 0.005, 0.0015 and 0.005 µg/ml of anti-BDCA2 mAbs (Biogen Idec) or 10 µg/ml of an isotype control. The anti-BDCA2 mAb, AC144, was purchased from Miltenyi. All conditions were set up in duplicate. Cells were stimulated with a final concentration of 1 µM of CpG-A or 50 µg/ml of poly(I:C) (Invivogen) and cultured for 16 h at 37°C and 5% $CO_2$. The supernatants were collected, and IFNα was evaluated.

#### *Isolated pDCs*

PDCs were isolated as previously described. Cells were plated at $1 \times 10^5$ cells/well in complete RPMI media in 96-well round-bottom plates. Cells were treated with 10, 1, 0.1, 0.001 and 0.0001 µg/ml anti-BDCA2 antibodies 24F4A, 24F4A-ef, Chi-6G6 (Biogen Idec) and AC144. As controls, pDCs were also stimulated in the presence of 10 µg/ml of an isotype control or 10 µg/ml of a fully humanized anti-CD40 antibody (Biogen). All conditions were set up in duplicate. PDCs were stimulated with either 1 µM of CpG-A or 5 µM or R848 (Invivogen). Cells were cultured for 16 h at 37°C and 5% $CO_2$. Supernatants were collected, and IFNα was evaluated.

#### *IFNα ELISA*

All cultured supernatants were evaluated for IFNα using VeriKine TM Human Interferon Alpha Serum Sample ELISA kit (PBL Interferon Source). The average IFNα levels of duplicate wells are calculated and depicted in the representative plots. The individual data points are shown in Supplementary Fig S11.

### Immune complex stimulation of pDCs

Anti-RNP antibodies were reconstituted in 2 ml of water. For each sample tested, immune complexes (IC) were pre-formed by mixing 1.25 µl Sm/RNP antigen (Arotec) and 2.5 µl anti-RNP antibodies (RayBiotech). The Sm/RNP IC mixture was incubated for 30 min at room temperature. The mixture was then diluted in 46.75 µl of media and added to 200 µl of cells in RPMI media. Cells were treated with anti-BDCA2 mAbs as previously described or with 10 µg/ml of anti-CD32 (AT10, AbD Serotec).

### *In vitro* internalization assays in whole blood and isolated pDCs

Whole blood was treated with 10, 3.33, 1.11, 0.37, 0.124, 0.04, 0.014, 0.005, 0.0015 and 0.005 µg/ml of 24F4A or 10 µg/ml 24F4A

for various amounts of time at 37°C and 5% $CO_2$. Blood was then stained with anti-CD123 (7G3), anti-CD20 (2H7), anti-CD14 (M5E2), anti-HLADR (L243) (BD Biosciences) and anti-BDCA2, clone 2D6 (non-cross-blocking Biogen Idec). Blood was lysed using BD lyse/fix solution (BD Biosciences) following manufacturer's protocol. PDCs were defined as $CD14^- CD20^- HLA\text{-}DR^+ CD123^+$ cells (Grouard *et al*, 1997). Flow cytometry was used to evaluate BDCA2 levels on pDCs. Isolated pDCs were treated with 10, 1, 0.1, 0.001 and 0.0001 µg/ml anti-BDCA2 antibodies, anti-CD40 antibody (Biogen) or an isotype control (human IgG1, Biogen). Cells were stimulated with 1 µM of CpG-A and cultured for 16 h at 37°C and 5% $CO_2$. To determine cell surface BDCA2 and CD32, pDCs were stained with anti-BDCA2, clone 2D6 (Biogen), and anti-CD32 (AT10, Life Technologies). Cells were acquired on a LSRII or Fortessa and analyzed in FlowJo.

### Intracellular signaling

Isolated pDCs were incubated for 10 min at 37°C and 5% $CO_2$ in the presence of 10 µg/ml of 24F4A, 24F4A-ef, or the isotype control. Cells were fixed with 2% PFA, permeabilized with methanol and stained with anti pPLCγ2 clone K86-689.37 (BD Biosciences) and anti pSYK clone C87C1 (Cell Signaling).

### Administration of 24F4A to cynomolgus monkeys and blood collection

Nine cynomolgus monkeys were randomized into three dosing groups (3/group) that received a single intravenous (IV) injection of 10 mg/kg 24F4A, 1 mg/kg 24F4A or sodium citrate (vehicle control). Both drug administration and *ex vivo* analyses were blinded to the three groups. Injections were administered at Toxikon Inc., Bedford, MA. In all studies, the animals were male cynomolgus macaques (*Macaca fascicularis*) 7–8 years of age. Animals were housed in standard stainless steel cages per USDA and AAALACI regulations and standards, and the study was performed under the IACUC protocol 2012-R-11. Blood was collected at Time 0 (immediately prior to administration of dosing solutions) and at 1, 6 h post-administration of dosing solutions, and days 1, 2, 3, 7, 8, 10, 14, 17, 21, 24, 28, 31, 35, 38, 42, 45, 52, 59, 66, 73, 92, 140 and 168 post-administration.

### Detection of serum 24F4A by ELISA

Plates were coated with 1 µg/ml of a human BDCA2-murine IgG2a Fc fusion protein and then blocked with diluent buffer (10 mM HEPES pH 7.0, 150 mM NaCl, 10 mM $CaCl_2$, 0.05% Tween-20, 1% BSA). Serum samples were added to the plates and incubated for 2 h. A donkey anti-human Fc (Jackson ImmunoResearch, 704-035-098) was then added, and peroxidase activity was measured using TMB substrate. The reaction was stopped with 2 N sulfuric acid prior to measurement of absorbance at 450 nm.

### MxA bioassay

Whole blood was diluted 1:4 in RPMI media and stimulated with CpG-A (200 µg/ml) for 16 h at 37°C and 5% $CO_2$. Supernatants were collected and cultured overnight with A549 cells, which produce MxA in response to IFN-I. MxA was quantified in A549 lysates by

ELISA following a previously established protocol (Wadhwa *et al*, 2013). A recombinant IFN standard curve was run in the MxA assay and used to determine the IFN-I levels in the actual samples.

### FACS analysis of cynomolgus monkey blood pre- and post-24F4A administration *in vivo*

#### BDCA2 detection (direct method)

Baseline BDCA2 expression (pre-dose) and BDCA2 expression post-dose were measured in cynomolgus monkey whole blood using labeled 24F4A (direct method). Pre- and post-dose cynomolgus monkey whole-blood samples were stained with anti-CD123 (7G3), anti-CD20 (2H7), anti-CD14 (M5E2), and anti-HLADR (L243) (BD Biosciences) to detect pDCs and with A647-labeled 24F4A to detect BDCA2.

#### 24F4A detection (indirect method)

Bound 24F4A was detected using PE-labeled anti-human IgG1 mAb (indirect method). To assess maximal 24F4A binding to cynomolgus pDCs in pre-dose samples, blood was "spiked" with 10 µg/ml of 24F4A at 4°C. The blood was then lysed (Easy Lyse, Leinco Technologies) and stained with anti-CD123 (7G3), anti-CD20 (2H7), anti-CD14 (M5E2), and anti-HLADR (L243) (BD Biosciences) to detect pDCs, and anti-human IgG1-PE mAb (Life Technologies) to identify bound 24F4A. The same indirect staining method was used to detect 24F4A on pDCs in post-dose cynomolgus whole-blood samples. Cells were acquired on a LSRII and analyzed in FlowJo.

### Statistical analyses: *ex vivo* TLR9 (CpG-A)-induced type I IFN production in whole-blood assay from cynomolgus monkeys treated with 24F4A

Mann–Whitney two-tailed parametric test with Welch's correction was used to compare the levels of IFN-I production in different treatment conditions. SD was not assumed to be equal. $P \leq 0.05$ was considered to be significant. To compare IFN-I concentrations from IV 24F4A-treated cohort, we used a two-way mixed-effects analysis of variance (ANOVA) to fit to $\log_{10}$ values of IFN-I concentrations from samples obtained both prior to and including 31 days after intravenous dosing or the last day prior to loss of the pharmacodynamics effect (BDCA2 internalization). The model was parameterized to estimate the mean $\log_{10}$ concentrations and the post- versus pre-dose differences for each dose group. The model also included three sets of random effects, which partitioned the sources of variation due to individual cynomolgus monkeys, bleeding time points within cynomolgus monkeys, and samples within ELISA plates. The proportion of variation due to animal differences was 69% of the total variability, with the remainder being primarily due to differences between time points within cynomolgus monkey (26%), and a small amount (< 6%) due to assay sources of variation.

### Confocal imaging

Cells were seeded in fibronectin-pre-coated glass chambers and cultured overnight in the presence 10 ng/ml of IL-3. AF647-labeled 24F4A was added for varying times. Cells were fixed with 2% paraformaldehyde at room temperature for 30 min and blocked/

permeabilized with PBS containing 2% BSA and 0.2% Triton X-100 for 1 h at RT. Cells were stained with anti-human LAMP-1 (H4A3, Southern Biotech) and then stained with anti-mouse IgG Cy3. Cells were then treated with phalloidin-A488 (Life Technologies) for 30 min at room temperature to visualize cytoskeleton. Cells were mounted using Pro Long Gold with DAPI (Life Technologies). Confocal images were acquired using a spinning disk confocal head (CSU-W1, PerkinElmer, Boston, MA) coupled to a fully motorized inverted Zeiss Axio Observer microscope (Carl Zeiss, Jena, Germany) equipped with a 63× oil objective lens (Pan Apochromat, 1.4 numerical aperture) and with a LED light source (X-Cite XLED1; Spectra Services, Ontario, NY). The imaging system operated under control of SlideBook 6 (Intelligent Imaging Innovations, Denver, CO). Image channels were adjusted within SlideBook and exported to Adobe Photoshop, which was used to create the figures.

### Statistical analysis

Statistical analysis for cynomolgus studies was done as described above. For all non-cynomolgus studies, the mean and standard deviation were calculated using PRISM 6.0. All experiments were repeated at least three times.

**Supplementary information** for this article is available online: http://embomolmed.embopress.org

### Acknowledgements

We thank Drs. Ellen Garber and Justin Caravella for mAb humanization; Diana Velez for mAb screening; Dr. Xiangyang Tan, Robert Dunn, and Weixing Yang for mAb generation; Chioma Nwankwo, You Li, Ben Snawder, Holly Legault, and Jeff Bajko for technical help; Drs. Carrie Wager and Wenting Wang for statistical analyses; Janine Ferrant-Orgettas for FcR binding assays and helpful suggestions; Drs. Agnes Gardet, Christine Loh, and Cherie Butts for review of manuscript; and Dr. Tomas Kirchhausen for analyses of confocal studies.

**The paper explained**

**Problem**

Systemic lupus erythematosus (SLE) is an autoimmune disease with high unmet medical need. Type I IFN derived from plasmacytoid dendritic cells (pDCs) is thought to play an important role in the pathogenesis of SLE. BDCA2 is a receptor that is specifically expressed on pDCs, and when engaged, it can inhibit type I IFN production from stimulated pDCs.

**Results**

We have generated an anti-BDCA2 antibody, 24F4A, that can internalize BDCA2 and inhibit type I IFN from stimulated pDCs *in vitro* (human) and *in vivo* (cynomolgus monkeys). In addition, we have identified a dual mechanism by which 24F4A can inhibit pDCs stimulated with pathogenic SLE autoantibodies.

**Impact**

In this study, we present a specific approach for targeting pDCs and inhibiting their ability to produce type I IFN. We uncovered additional mechanisms where an anti-BDCA2 antibody could have added efficacy and better safety in SLE.

## Author contributions

AP established hypotheses, designed and performed experiments, analyzed data, wrote parts of the paper, and edited the paper; KO performed confocal studies, designed experiments, and edited the paper; JC designed and performed experiments; HKC designed and performed experiments and edited the paper; RIS designed and performed experiments; AR designed experiments; KLO designed experiments and edited the paper; FRT established hypotheses designed and performed experiments; TC established hypotheses, designed and performed experiments, and edited the paper; JLV established hypotheses and edited the paper; and DR established hypotheses, designed experiments, analyzed data, and wrote the paper. All authors critically reviewed the paper.

## Conflict of interest

The authors are employees of Biogen Idec.

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
