## [Review Process File · EMBO Molecular Medicine]

Manuscript EMM-2014-04719

Anti-BDCA2 monoclonal antibody inhibits plasmacytoid dendritic cell activation through Fc-dependent and independent mechanisms

Alex Pellerin, Karel Otero, Julie M. Czerkowicz, Hannah M Kerns, Renée I. Shapiro, Ann Ranger, Kevin L. Otipoby, Frederick R. Taylor, Thomas Cameron, Joanne L. Viney, and Dania Rabah

Corresponding author: Dania Rabah, Biogen Idec

Review timeline:

Submission date:	02 October 2014
Editorial Decision:	28 October 2015
Revision received:	23 December 2014
Editorial Decision:	22 January 2015
Revision received:	02 February 2015
Accepted:	15 February 2015

Transaction Report:

Editor: Céline Carret

1st Editorial Decision

28 October 2015

Thank you for the submission of your manuscript to EMBO Molecular Medicine. We have now heard back from the three referees whom we asked to evaluate your manuscript. Although the referees find the study to be of potential interest, they also raise a number of concerns that need to be fully addressed in the next version of your article.

As you will see from the comments below, while referee 2 is rather enthusiastic about the study, the other two referees, while appreciating the importance of the findings are much more reserved. Referee #3 raised serious technical issues that reduce the overall significance of the findings. And referee #1 is concerned about the novelty. Therefore, we would strongly encourage you to address all the technical problems highlighted by referee #3 and confirmed by referee 1 in a separate e-mail, who agreed that comparing anti-BDCA-2 mAb to other ones like AC144 would be extremely helpful, as well as understanding whether CD32b is expressed by PDCs and what is CD40 contribution to CD32 internalisation. In addition, referees recommend detailing and clarifying here and there, adding references too, and better discussing the findings to highlight the novel contribution of this study to the scientific community.

Given the balance of these evaluations, we would like to invite a revision of your manuscript with the understanding that all the issues raised have to be addressed in a satisfactory manner. Please note that it is EMBO Molecular Medicine policy to allow only a single round of revision and that, as acceptance or rejection of the manuscript will depend on another round of review, your responses should be as complete as possible.

I look forward to receiving your revised manuscript.

***** Reviewer's comments *****

Referee #1 (Comments on Novelty/Model System):

I think most of the results are confirmatory in nature:

Figure 1: essentially confirmatory to previous findings

Figure 2: essentially confirmatory to previous findings

Figure 3: first study in cynomolgus monkeys, but another in vivo model has been published before (not mentioned at all).

Figure 4: basically the only really new finding: wildtype IgG1 anti BDCA-2 is more potent than Fc-effectorless anti BDCA-2 at inhibiting type I interferon induction by immune complexes. This may be due to additional down-modulation of CD32a on PDCs.

Referee #1 (Remarks):

More than 13 years ago Dzionek et al. (1) in an original article published in the Journal of Experimental Medicine and Rⁿⁿblom et al. (2) in a comment on the article reported that BDCA-2-ligation on PDCs with a specific mAb suppresses induction of type I interferon expression in PDCs and may therefore be an effective therapy for the treatment of SLE.

Here, Pellerin and coworkers demonstrate that intravenous injection of a humanized anti BDCA-2 antibody in cynomolgus monkeys inhibits the ability of circulating PDCs to produce type I interferon. Furthermore, the authors show that human IgG1 wildtype anti BDCA-2 mAb is more potent than human IgG4.P/human IgG1 chimeric anti BDCA-2 mAb at inhibiting Sm/RNP immune complex-induced type I interferon production in PDCs. Based on the observation that BDCA-2 mAb-ligation leads to concurrent internalization of BDCA-2 and CD32, the authors claim that IgG1 wildtype anti BDCA-2 mAb is more potent due to additional down-modulation (depletion) of CD32a, the uptake receptor for stimulatory immune-complexes.

I think the results are mostly confirmatory in nature:

1. Figure 1: BDCA-2 ligation by the fully humanized IgG1 mAb 24F4A inhibits induction of type I interferon production in PDCs by CPG-A (dose-dependent, IC₅₀ of 0.04 +/- 0.05 g/mL, whole blood as well as PBMC, healthy blood donors as well as SLE patients)

Dzionek et al. (1) have already shown in 2001 that BDCA-2 ligation by the murine IgG1 mAb AC144 inhibits induction of type I interferon production in PDCs by various stimuli (dose-dependent, IC₅₀ of < 0.1 g/mL, PBMC, healthy blood donors). Blomberg et al. (3) have already demonstrated in 2003 that the same is true for PDCs from SLE patients.

2. Figure 1 and 2: BDCA-2 ligation by the fully humanized IgG1 mAb 24F4A induces rapid internalization of BDCA-2 (dose-dependent, EC₅₀ of 0.017 {plus minus} 0.005 g/mL, correlation

with IC50 for inhibition of type I interferon production, R2 of 0.68, whole blood; dose-dependent internalization kinetics, no receptor recycling to the cell surface, co-localization with LAMP1 after 6 hrs)

Dzionic et al. (4) have already shown in 2000 that BDCA-2-ligation by the murine IgG1 mAb AC144 results in rapid receptor-mediated endocytosis of the mAb. Jaehn et al. (5) have already demonstrated in 2008 that the internalized anti BDCA-2 mAb traffics via early EEA1+ sorting endosomes to LAMP1+ lysosomes.

3. Figure 3: PDCs from cynomolgus monkeys treated with the fully humanized IgG1 mAb 24F4A (1 mg/kg, 10 mg/kg) produce less type I interferon after CpG-A stimulation in vitro than PDCs from vehicle-treated animals (1 mg/kg: 95% CI: 18%-65%; p=0.004).

In a xenograft model of human psoriasis (uninvolved prepsoriatic skin of psoriasis patients spontaneously converts into a full-fledged psoriatic skin lesion on transplantation onto AGR / mice within 35 d), Nestle et al. (6) have already shown in 2005 that intravenous injections of the murine anti BDCA-2 mAb AC144 blocked the production of type I interferon by PDCs in engrafted skin and inhibited the development of a psoriatic phenotype. These preexisting in vivo data are not mentioned at all in the manuscript from Pellerin et al..

4. Figure 4 and 5: Comparison of wt IgG1 24F4A and Fc-effectorless human IgG4.P/human IgG1 chimeric 24F4A (24F4A-ef):

- a) both forms are equipotent at inhibiting induction of type I interferon production in PDCs by R848 (TLR7-ligand) and CpG-A (TLR9 ligand), but 24F4A is 40-fold more potent than 24F4A-ef at inhibiting Sm/RNP immune complex-induced type I interferon production in PDCs.
- b) 24F4A and 24F4A-ef are equipotent at inducing BDCA-2 internalization, but only 24F4A concurrently down-modulates CD32 cell surface expression on PDCs (not true for CD40 mAb with same Fc region).

To my mind, Figure 4 shows the only really new findings.

Additional major criticisms:

1. The authors conclude that IgG1 wildtype anti BDCA-2 mAb is more potent than Fc-effectorless anti BDCA-2 mAb due to additional down-modulation (depletion) of CD32a, the uptake receptor for stimulatory immune-complexes. It is not clear to me, whether PDCs express CD32b and whether AT10 recognizes CD32b.

If CD32b is expressed by PDCs (and AT10 recognizes CD32b), IgG1 wildtype anti BDCA-2 mAb may be more potent than Fc-effectorless anti BDCA-2 mAb due to CD32b binding and inhibitory signaling. In this case, CD32 down-modulation (depletion) may be an epiphenomenon.

Additional minor criticisms:

- 2. For me Figure 3A iii and iv look pretty similar (vehicle vs treated).
- 3. Humanization of 24F4A by CDR grafting is not described with adequate precision.

1. Dzionic A, Sohma Y, Nagafune J, Cella M, Colonna M, Facchetti F, Gunther G, Johnston I, Lanzavecchia A, Nagasaka T, Okada T, Vermi W, Winkels G, Yamamoto T, Zysk M, Yamaguchi Y, Schmitz J. 2001. BDCA-2, a novel plasmacytoid dendritic cell-specific type II C-type lectin, mediates antigen capture and is a potent inhibitor of interferon alpha/beta induction. *J Exp Med* 194: 1823-34.

2. Ronnblom L, Alm GV. 2001. A pivotal role for the natural interferon alpha-producing cells (plasmacytoid dendritic cells) in the pathogenesis of lupus. *J Exp Med* 194: F59-63

3. Blomberg S, Eloranta ML, Magnusson M, Alm GV, Ronnblom L. 2003. Expression of the markers BDCA-2 and BDCA-4 and production of interferon-alpha by plasmacytoid dendritic cells in systemic lupus erythematosus. *Arthritis Rheum* 48: 2524-32

4. Dzionic A, Fuchs A, Schmidt P, Cremer S, Zysk M, Miltenyi S, Buck DW, Schmitz J. 2000.

BDCA-2, BDCA-3, and BDCA-4: three markers for distinct subsets of dendritic cells in human peripheral blood. *J Immunol* 165: 6037-46.

5. Jaehn PS, Zaenker KS, Schmitz J, Dzionek A. 2008. Functional dichotomy of plasmacytoid dendritic cells: antigen-specific activation of T cells versus production of type I interferon. *Eur J Immunol* 38: 1822-32

6. Nestle FO, Conrad C, Tun-Kyi A, Homey B, Gombert M, Boyman O, Burg G, Liu YJ, Gilliet M. 2005. Plasmacytoid predendritic cells initiate psoriasis through interferon-alpha production. *J Exp Med* 202: 135-43

Referee #2 (Remarks):

The manuscript by Rabah and colleagues describes the generation of a new mAb against BDCA-2, an inhibitory receptor on human plasmacytoid dendritic cells (pDCs). BDCA-2 is highly specific for pDCs and is the obvious first choice for therapeutic targeting of pDCs; inexplicably, however, this has never been attempted. The authors verify the ability of the new mAb to inhibit IFN production by pDCs in humans and non-human primates, as well as describe an important role of the Fc fragment in the blockade of pDC activation by immune complexes, a potentially relevant ligand in autoimmunity. This work is the first focused attempt at therapeutic targeting of pDCs, and the first essential step in this direction. As such, it is novel, timely, important and of broad interest to the field.

Major comment:

Additional characterization of the mAb should be presented to support its stated specificity for BDCA-2: binding to the recombinant protein and/or to BDCA-2 transfected cells versus controls, co-staining with other pDC markers in PBMC etc. Of note, only a fraction of CD123hi HLA-DR+ PBMC are positive by FACS (Fig. S2) - because BDCA-2 is supposedly expressed on all pDCs, this discrepancy should be resolved, e.g. by more specific gating.

Minor comment:

in cynomolgus monkeys, the mAb appears to inhibit pDC function without depleting pDCs. This result is potentially very important, suggesting that the mAb would work through functional modulation rather than simple cell depletion. In that respect, Sisirak et al. (*JEM* 2014) showed that even a partial functional impairment of pDCs drastically improves lupus in mice; it may be useful to emphasize and discuss the parallels between the studies. Furthermore, complete pDC depletion impairs control of persistent viral infections (Cervantes-Barragan et al., *PNAS* 2012), whereas a partial functional modulation would be less immunosuppressive. The authors may consider discussing this potentially advantageous aspect of their approach.

Referee #3 (Remarks):

The manuscript 'Anti-BDCA2 monoclonal antibody inhibits plasmacytoid dendritic cell activation through Fc-dependent and independent mechanisms' by Pellerin et al represents a putatively interesting paper. However, the sometimes poor description of experiments and figures dampens my enthusiasm. Here are my comments:

- a panel of antibodies is developed against the pDC receptor BDCA2. One of these antibodies is selected: 24F4A. On basis of what? How many were generated, what were their isotypes, what epitope do they recognize, what is their affinity? Octet analysis was mentioned somewhere, but nowhere shown. Also the humanization of this antibody to a human IgG1 is poorly described. It would be nice in figure 1A to have at least an comparison with other BDCA2 antibodies, if not from the panel then at least the 2D6 antibody that was mentioned later, and also the antibody AC144 (mouse IgG1 from Miltenyi). Also in the other assays for internalization of BDCA2 en FcγRIIa these other antibodies should be included, because it is interesting to know how specific the

internalization is for this antibody or more like a general mechanism.

I assume that the antibody is (being) patented, is this why not much information is given? And should the patent, when existing, not be mentioned in the text?

In general it is hard to interpret the figures, providing clearer legends would help.

Specific comments:

M&M:

- antibody production: was this done in serum-free medium?
- Please replace cynos with Cynomolgus Monkeys or Macaques out of respect for the animal. Please be consistent in cynomolgus, not cynomologous or cynomolgous. What does 'intact male Cynomolgus Macaques mean? And what are squeeze cages? Doesn't sound very friendly...

Figures:

- Fig1 1A: an average is shown of duplicate, this is of course not possible, you need at least triplicates to show averages with standard deviation. In the case of on duplicate, the separate graphs should be shown.

- Fig 1B: IC50 of 0.04 +/- 0.05, how is this possible? Negative amount of antibody can still inhibit IFN production?

- Fig 1ACD, 5ACF: please avoid negative numbering of the X-axis.

- Fig 2B: no recycling, after 16 hrs still no BDCA2 on surface. Is there no de novo synthesis? Is the turn over of this protein so extremely low that in 16 hrs no new protein is produced? Please explain.

- Fig2D no co-localization between LAMPI and BDCA2 is shown, although the text says: persists to 6 hrs, suggesting it is already there at 10 minutes, which is not the case. In Fig 2E the cells look really strange, please show cells with better morphology. The text suggest it co-localizes with TLR9, please show this.

- Figure 3a is really hard to understand. Was the indirect and direct detections always combined? I understand there is internalization of BDCA2 after 6 hrs antibody treatment in vivo, but shouldn't the dotted line in fig3Aiii be negative, as in vehicle control with indirect label? Please clarify.

- there is no figure 3B-v, but it is referred to in the text

- please explain a bit more on MxA bioassay.

- typo in sentence: MxA bioassay(..) that indirectly measure... should be measures

Figure 3C: the decrease of IFN levels in the monkeys is not really impressive. What happened to Cyno A07858? Measurement stopped on day 14, but in figure 3Bi there are still data after this day.

- It is interesting to see the effect on FcγRIIa. The authors claim this is the only Fc receptor of pDC, but they also express IIb (see review nature immunology 2014). It is hard to discriminate between IIa and b.

- Fig 4C: legend does not match with X-axis (CPG vs R848)

- The authors show that only with immunocomplexes and the BDCA2 antibody the FcR disappears. But how were the IC prepared? The M&M is not clear on this, what is the ratio between antibody and antigen?

- it is good that they do not only use an isotype control but also an CD40 antibody that can bind both with Fab and Fc part. This control antibody does not trigger internalization of FcR. However, this could be due to the expression level of the target, are BDCA2 and CD40 equally expressed? QIFI analysis of pDC would show this.

- again, include the other BDCA2 antibodies for analyses of FcR internalization.

In the discussion, the authors mention that it was impossible to show co-localisation of CD32a and BDCA2. However, confocal picture of CD32a has been shown more often, so this should be possible and added to the paper. Also, does the same mechanism of CD32a internalization happens in cynomolgus monkeys?

Please include reference for 'scorpion' mechanism with first sentence. Isn't this the same as described by Kurlander a long time ago?

References: I think the nice overview of IFN directed treatment of SLE by Kirou in clinical Immunology of 2013 should be included.

Referee #1:

We sincerely appreciate the comments raised by the referee, including those regarding the novelty of our work. With the referee's comments in mind we have critically revised the manuscript to differentiate the novelty of our results as compared to what has been previously published, which we believe has improved the message of our manuscript.

1. Figure 1: BDCA-2 ligation by the fully humanized IgG1 mAb 24F4A inhibits induction of type I interferon production in PDCs by CPG-A (dose-dependent, IC50 of 0.04 +/- 0.05 µg/mL, whole blood as well as PBMC, healthy blood donors as well as SLE patients)

Dzionic et al. (1) have already shown in 2001 that BDCA-2 ligation by the murine IgG1 mAb AC144 inhibits induction of type I interferon production in PDCs by various stimuli (dose-dependent, IC50 of < 0.1 µg/mL, PBMC, healthy blood donors). Blomberg et al. (3) have already demonstrated in 2003 that the same is true for PDCs from SLE patients.

We agree that Figure 1, panels A, B and C are confirmatory to previously published data. However, we needed to show that we recapitulate previously reported data with our newly generated anti-BDCA2 mAb, 24F4A. Figure 1 extended those results to show that 24F4A is capable of inhibiting TLR9-induced IFN-I and mediating BDCA2 internalization in whole blood assays. We confirmed that pDCs were the only producers of TLR9-induced IFN-I in those cultures using intracellular staining (our unpublished observations). For the first time, we show the ability to correlate BDCA2 engagement and internalization with the inhibition of IFN-I.

With the referee's comments in mind we modified Figure 1 to:

1. *Clearly re-state published data on the effect of the anti-BDCA2 mAb, AC144, on pDCs in the Result and Discussion sections.*
2. *Include details on our mAb generation effort and how it compares with AC144. This is now depicted in Supplementary Table 1 and Figure S1.*
3. *Highlight the novelty of the observed link between BDCA2 internalization and IFN-I inhibition.*

2. Figure 1 and 2: BDCA-2 ligation by the fully humanized IgG1 mAb 24F4A induces rapid internalization of BDCA-2 (dose-dependent, EC50 of 0.017 ± 0.005 µg/mL, correlation with IC50 for inhibition of type I interferon production, R2 of 0.68, whole blood; dose-dependent internalization kinetics, no receptor recycling to the cell surface, co-localization with LAMP1 after 6 hrs)

Dzionic et al. (4) have already shown in 2000 that BDCA-2-ligation by the murine IgG1 mAb AC144 results in rapid receptor-mediated endocytosis of the mAb. Jaehn et al. (5) have already demonstrated in 2008 that the internalized anti BDCA-2 mAb traffics via early EEA1+ sorting endosomes to LAMP1+ lysosomes.

The aim of the experiments shown in Figure 2 is to further interrogate the link between the anti-BDCA2-mediated BDCA2 internalization and IFN-I inhibition.

The novelty of the results shown in this figure is 3 fold:

1. *It provides the first direct evidence that BDCA2 does not recycle back to the cell surface but persists in the endolysosomal compartment (Figure 2C).*
2. *It provides evidence that a 1-hour pre-incubation with 24F4A, which mediates complete internalization of BDCA2, is sufficient to inhibit TLR9-induced IFN-I production (Figure 2D).*
3. *We also extended the pre-incubation studies and we ascertained that BDCA2 is still functional over longer pre-incubation periods inside intracellular compartments. As shown in Figure 2E, pre-incubation with 24F4A for up to 9 hours was still capable of inhibiting TLR9-induced IFN-I production.*

Our data may suggest that the persistent localization of the 24F4A/BDCA2 complex in LAMP1⁺ endolysosomal compartment in the absence of recycling could be important for its ability to mediate IFN-I inhibition.

To highlight the novelty of the findings in Figure 2 we:

1. *Clearly cited previous published results.*
2. *We changed the order of the panels to highlight the most novel findings which are the lack of BDCA2 recycling and the impact of intracellular localization of BDCA2 on IFN-I inhibition.*
3. *We included a new experiment that shows that internalized BDCA2 is still capable of mediating IFN-I inhibition over prolonged pre-incubation periods with 24F4A (Figure 2E).*

3. Figure 3: PDCs from cynomolgus monkeys treated with the fully humanized IgG1 mAb 24F4A (1 mg/kg, 10 mg/kg) produce less type I interferon after CpG-A stimulation in vitro than PDCs from vehicle-treated animals (1 mg/kg: 95% CI: 18%-65%; p=0.004).

In a xenograft model of human psoriasis (uninvolved prepsoriatic skin of psoriasis patients spontaneously converts into a full-fledged psoriatic skin lesion on transplantation onto AGR^h mice within 35 d), Nestle et al. (6) have already shown in 2005 that intravenous injections of the murine anti BDCA-2 mAb AC144 blocked the production of type I interferon by PDCs in engrafted skin and inhibited the development of a psoriatic phenotype. These preexisting in vivo data are not mentioned at all in the manuscript from Pellerin et al..

We agree that the paper by Nestle et. al. should have been cited, and we apologize for our oversight.

We have now revised the discussion to clearly highlight the novelty and implication of our in vivo data using the cynomolgus monkey system:

1. *We show for the first time that an effector competent anti-BDCA2 mAb can induce BDCA2 internalization in vivo and does not lead to pDC depletion.*
2. *The study demonstrates the ability of the anti-BDCA2 mAb to dampen IFN-I production in vivo in cynomolgus monkeys.*

The anti-BDCA2-mediated BDCA2 internalization and IFN-I inhibition in the absence of pDC depletion in vivo suggest that a treatment with our antibody could be an advantageous approach to mediate therapeutic efficacy in autoimmune and inflammatory diseases without the potential immunosuppressive effects of pDC depletion.

4. Figure 4 and 5: Comparison of wt IgG1 24F4A and Fc-effectorless human IgG4.P/human IgG1 chimeric 24F4A (24F4A-ef):

a) both forms are equipotent at inhibiting induction of type I interferon production in PDCs by R848 (TLR7-ligand) and CpG-A (TLR9 ligand), but 24F4A is 40-fold more potent than 24F4A-ef at inhibiting Sm/RNP immune complex-induced type I interferon production in PDCs.

b) 24F4A and 24F4A-ef are equipotent at inducing BDCA-2 internalization, but only 24F4A concurrently down-modulates CD32 cell surface expression on PDCs (not true for CD40 mAb with same Fc region).

To my mind, Figure 4 shows the only really new findings.

As the referee appreciates, Figures 4 and 5 describe a novel mechanism by which anti-BDCA2 mAb can inhibit SLE-IC-mediated pDC activation. In addition to IFN-I inhibition mediated by ligation of BDCA2 and its intracellular localization, the second mechanism is mediated by the depletion of Fc receptor (FcR) from the surface of pDCs. To further support our conclusions, in the revised manuscript we have now included a comparison with another mAb (6G6) from our anti-BDCA2 antibody panel that we engineered to have a human Fc region identical to that of 24F4A. This mAb leads to marginal BDCA2 internalization and similarly marginal CD32a internalization (Figure 5). We also provide evidence that another effector competent anti-BDCA2 mAb, clone AC144, is capable of mediating CD32a internalization. These data are presented in Supplementary Figure S9. Together, we unequivocally demonstrate that anti-BDCA2 mAbs capable of driving BDCA2 internalization cause CD32a disappearance from the cell surface, suggesting that CD32a internalization is driven by BDCA2 internalization.

Additional major criticisms:

The authors conclude that IgG1 wildtype anti BDCA-2 mAb is more potent than Fc-effectorless anti BDCA-2 mAb due to additional down-modulation (depletion) of CD32a, the uptake receptor for stimulatory immune-complexes. It is not clear to me, whether PDCs express CD32b and whether AT10 recognizes CD32b.

If CD32b is expressed by PDCs (and AT10 recognizes CD32b), IgG1 wildtype anti BDCA-2 mAb may be more potent than Fc-effectorless anti BDCA-2 mAb due to CD32b binding and inhibitory signaling. In this case, CD32 down-modulation (depletion) may be an epiphenomenon.

We thank the referee for raising this question. We have confirmed the exclusive expression of CD32a vs. CD32b that had previously been reported (Bave et al, 2003) using Q-PCR. These data are currently presented in Figure S8. Based on these results, we are confident that anti-CD32 mAb, clone AT10, is strictly identifying CD32a on the surface of pDCs. As the referee mentioned, CD32b engagement on B cells has been shown to inhibit signaling downstream from the B cell receptor. Therefore CD32b engagement is expected to affect the BDCA2-mediated BCR-like signaling cascade. CD32b engagement and impact on BDCA2 signaling would not have discriminated between immune complex-stimulation and stimulation with synthetic ligands. The fact that the added potency of 24F4A effector competent mAb is only seen with immune complex stimulation confirms that it is strictly mediated by the depletion of CD32a from the surface of pDCs.

Additional minor criticisms:

For me Figure 3A iii and iv look pretty similar (vehicle vs treated).

Figure 3A iii in the original submission shows that there is no 24F4A detectable on the surface of pDC taken from vehicle treated animals, yet there is BDCA2 expression on these cells (Figure 3A i). This is expected, because animals were not dosed with 24F4A.

Figure 3A iv shows that there is also no 24F4A detectable on the surface of pDC taken from 24F4A treated animals, AND there is no BDCA2 expression on these cells (Figure 3A ii). The lack of detectable 24F4A, and the lack of BDCA2, suggests BDCA2 has internalized following 24F4A administration.

We have now revised Figure 3A for a more clear depiction of these results.

Humanization of 24F4A by CDR grafting is not described with adequate precision.

We have included a detailed description of the humanization of 24F4 in the Supplementary Materials and Methods section.

Referee #2 (Remarks):

The manuscript by Rabah and colleagues describes the generation of a new mAb against BDCA-2, an inhibitory receptor on human plasmacytoid dendritic cells (pDCs). BDCA-2 is highly specific for pDCs and is the obvious first choice for therapeutic targeting of pDCs; inexplicably, however, this has never been attempted. The authors verify the ability of the new mAb to inhibit IFN production by pDCs in humans and non-human primates, as well as describe an important role of the Fc fragment in the blockade of pDC activation by immune complexes, a potentially relevant ligand in autoimmunity. This work is the first focused attempt at therapeutic targeting of pDCs, and the first essential step in this direction. As such, it is novel, timely, important and of broad interest to the field.

We thank the referee for the thorough review of the manuscript and for the enthusiastic support of our finding. We are excited about the potential of 24F4A as a therapeutic to dampen the IFN-I response in a disease setting. We are also excited to report an additional mechanism of action for this antibody that may improve its clinical efficacy.

Major comment:

Additional characterization of the mAb should be presented to support its stated specificity for BDCA-2: binding to the recombinant protein and/or to BDCA-2 transfected cells versus controls, co-staining with other pDC markers in PBMC etc. Of note, only a fraction of CD123^{hi} HLA-DR⁺ PBMC are positive by FACS (Fig. S2) - because BDCA-2 is supposedly expressed on all pDCs, this discrepancy should be resolved, e.g. by more specific gating.

We thank the referee for these comments. We agree that BDCA2 is specifically expressed on all pDCs. We believe the figure was confusing in its depiction of 24F4A and 2D6 co-staining of CD123⁺ and HLADR⁺ pDCs. In the original Figure S2, the staining for BDCA2 with labeled 24F4A and the non-cross blocking mAb 2D6 was shown in red in the dot plot, while the FMO for both antibodies was shown in blue. We believe this led to the confusion that 24F4A and 2D6 were not staining all pDCs. We have revised the figure to only represent BDCA2 staining on CD123⁺, HLADR⁺ cells. To the referee's point, all CD123⁺ HLA-DR⁺ cells stain positive for BDCA2. In addition, we have added a supplementary figure (Figure S5) that shows octet binding of 24F4A to human and cynomolgus BDCA2 as well as specific binding to native BDCA2 on pDCs in human and cynomolgus monkey.

Minor comment:

In cynomolgus monkeys, the mAb appears to inhibit pDC function without depleting pDCs. This result is potentially very important, suggesting that the mAb would work through functional modulation rather than simple cell depletion. In that respect, Sisirak et al. (JEM 2014) showed that even a partial functional impairment of pDCs drastically improves lupus in mice; it may be useful to emphasize and discuss the parallels between the studies. Furthermore, complete pDC depletion impairs control of persistent viral infections (Cervantes-Barragan et al., PNAS 2012), whereas a partial functional modulation would be less immunosuppressive. The authors may consider discussing this potentially advantageous aspect of their approach.

We thank the referee for these insightful comments. We completely agree that the potential of the anti-BDCA2 antibody to inhibit pDC without depleting them is very important for its therapeutic application, and this point is now developed in the discussion section. Your suggestion helped us highlight the novelty of this finding and the significance/uniqueness of our approach.

Referee #3 (Remarks):

The manuscript 'Anti-BDCA2 monoclonal antibody inhibits plasmacytoid dendritic cell activation through Fc-dependent and independent mechanisms' by Pellerin et al represents a putatively interesting paper. However, the sometimes-poor description of experiments and figures dampens my enthusiasm.

We thank the referee for the thorough review of the manuscript and the useful suggestions.

We have critically reviewed our manuscript to improve the description of the experiments when necessary. As a result, some figures, figure legends and description of the results have been revised.

Here are my comments:

- a panel of antibodies is developed against the pDC receptor BDCA2. One of these antibodies is selected: 24F4A. On basis of what? How many were generated, what were their isotypes, what epitope do they recognize, what is their affinity? Octet analysis was mentioned somewhere, but nowhere shown.

We agree that it is useful to show these data. Additional details on mAb generation have been added to the "mAb generation" section in the Supplementary Materials and Methods section. We present a description of the approach used for mAb generation, screening criteria, and affinity measurements by Octet. We added one supplementary table (Table 1) that lists all the mAbs that were prioritized based on the screening strategy and the rationale for choosing 24F4. We also included the data pertaining to 24F4A in Figure S5: Octet data, and binding curves to native BDCA2 on human and cynomolgus pDCs.

Also the humanization of this antibody to a human IgG1 is poorly described.

We have included a detailed description of the humanization of 24F4 in the Supplementary Materials and Methods section.

It would be nice in figure 1A to have at least an comparison with other BDCA2 antibodies, if not from the panel then at least the 2D6 antibody that was mentioned later, and also the antibody AC144 (mouse IgG1 from Miltenyi).

To address this suggestion we included additional experiments presented in a supplementary Figure in which we compare 24F4 to other mAbs; 15F3 from our panel and the commercial anti-BDCA2 mAb AC144 (Figure S1). We also included comparison of 24F4 to another antibody from our panel, 6G6 that has a different biology (Figure S4). 6G6 binds to a different epitope than that of 24F4A and the commercial antibody AC144 (Supplementary Table 1).

Also in the other assays for internalization of BDCA2 on FcγRIIa these other antibodies should be included, because it is interesting to know how specific the internalization is for this antibody or more like a general mechanism.

We agree with the referee that it is important to compare 24F4A to other anti-BDCA2 mAbs and understand whether all effector competent anti-BDCA2 mAbs can downmodulate CD32a or if it is specific to 24F4A. To this end, we revised Figure 5 to include 6G6, a mAb from our panel that leads to modest BDCA2 internalization and IFN- γ inhibition (Figure S4). We engineered a chimera of 6G6 that has a human Fc region identical to that of 24F4A (Chi-6G6). As expected, chi-6G6 led only to marginal BDCA2 internalization and similarly marginal internalization of CD32a. In addition, we compared 24F4A to AC144, and we show that both mAbs are capable of internalizing BDCA2 and CD32a. These data are now presented in supplementary Figure S9.

We thank the referee for this suggestion as we believe it made our message much stronger. Adding the results with Chi-6G6 in addition to anti-CD40 mAb strengthened the notion that the effect is not mediated by a mere blockade of CD32a binding. Collective data from 24F4A, Chi-6G6 and AC144 suggested that only effector competent anti-BDCA2 mAbs capable of mediating BDCA2 internalization could lead to CD32a downmodulation. This confirms that CD32a internalization is driven by BDCA2 internalization. These data lend more support to our previous proposal that other endocytic receptors on immune cells can lead to FcR depletion through a similar mechanism to that of BDCA2.

I assume that the antibody is (being) patented, is this why not much information is given? And should the patent, when existing, not be mentioned in the text?

Detailed information on the generation of the 24F4A antibody was left out to keep the manuscript within the recommended word count and focus on the biology of 24F4A. At the referee's suggestion, this information has been added in the Supplementary Materials and Methods and in Supplementary Table 1, Figure S1 and Figure S6. No patent has been granted yet on the anti-BDCA2 antibody described in the manuscript (24F4A). Accordingly, there is no patent information to provide at this time.

In general it is hard to interpret the figures, providing clearer legends would help.

We thank the referee for raising this concern. The legends have been revised to provide a more clear description of the experiments.

Specific comments:

M&M:

- Antibody production: was this done in serum-free medium?

The antibody generation was done in serum free medium, and it is now mentioned in mAb generation section in supplementary Materials and Methods.

- Please replace cynos with Cynomolgus Monkeys or Macaques out of respect for the animal. Please be consistent in cynomolgus, not cynomologous or cynomolgous.

We agree and have removed the cyno abbreviation. We refer to the cynomolgus monkeys used in this experiment with the correct spelling throughout the document.

What does 'intact male Cynomolgus Macaques mean? And what are squeeze cages? Doesn't sound very friendly...

Intact male Cynomolgus Macaques means that the cynomolgus monkeys are non-gonadectomized, non-castrated and chemically naïve.

Squeeze cages, are cages commonly used for temporarily restraining cynomolgus monkeys for IV injection which was done once. These cages have a movable back panel that slides forward to restrain the animal for the purpose of the success of the IV injection. This was done in a humane way was based on IACUC approved protocol.

Reference to "intact" and "squeeze cages" have been removed from the Materials and Methods text to avoid any confusion.

Figures:

- Fig1 1A: an average is shown of duplicate, this is of course not possible, you need at least triplicates to show averages with standard deviation. In the case of on duplicate, the separate graphs should be shown.

We agree with the referee that representing a standard deviation is not adequate since we have only 2 technical replicates. We added all the data points for all the IFN-I measurements in a supplementary figure (Figure S11). As shown in figure S11, the mean of the duplicates was representative of the individual curves. For simplicity, we presented the mean of the duplicates in the main figures and provided the reader with an estimate of the distance between the duplicates by showing the individual data points.

- Fig 1B: IC₅₀ of 0.04 +/- 0.05, how is this possible? Negative amount of antibody can still inhibit IFN production?

This large standard deviation is due to the right skewness of the IC₅₀ data (a couple of samples with extremely large IC₅₀ values). We have changed the analysis to a nonparametric approach in the revised manuscript for a more accurate representation of the data.

- Fig 1ACD, 5ACF: please avoid negative numbering of the X-axis.

The axes have been changed to anti-log in all the figures

- Fig 2B: no recycling, after 16 hrs still no BDCA2 on surface. Is there no de novo synthesis? Is the turn over of this protein so extremely low that in 16 hrs no new protein is produced? Please explain.

To address this question, we looked at the re-expression of BDCA2 by culturing the pDCs for longer periods of time (up to 72 h), in the presence of IL-3 to preserve cell viability. Isolated pDCs were pre-incubated in the absence or presence of 10 µg/mL of 24F4A for 1 h at 37°C to mediate maximal BDCA2 internalization. PDCs were then washed and incubated in the presence of IL-3 and BDCA2 receptor re-expression was measured by flow cytometry. As shown in Figure for Referees R1, in pDCs pre-treated with 24F4A (solid line) BDCA2 re-expression was at 30% 48 hours post-culture and 60% at 72 hours post-culture as compared to untreated cells (dotted lines).

Figure R1. Re-expression of BDCA2 after 24F4A treatment.

Isolated pDCs were incubated with 24F4A (black line) or PBS (dotted, gray line) for 1 hour at 37°C. Cells were then washed and cultured with 20ng/mL of IL-3 for the indicated amount of time. Flow cytometry was used to determine the levels of BDCA2 on the surface of pDCs using fluorescently labeled ant-BDCA2 (2D6).

While we agree with the referee that these results might be surprising, there are a few things to consider. First, pDCs are terminally differentiated cells in which protein turnover might be more modest than in a highly dividing cell type. Second, cell surface proteins can have half-lives of >72 h as shown by Hare and Taylor (Hare & Taylor, 1991). A protein with a half-life of 72 h would therefore only be synthesized at a corresponding replacement rate, and thus a lack of significant replacement within 24 h is not unexpected. Lastly, the retention of BDCA2 in intracellular compartments up to 16 hours and the continuous signaling and IFN-I inhibition mediated by BDCA2 ligation (now shown in figure 2 as panel E) might alter transcription in a way that reduces or eliminate BDCA2 protein surface expression, possibly as a “circuit breaker” to allow recovery or “resetting” of the cell to normal function after a certain period of time.

- Fig2D no co-localization between LAMP1 and BDCA2 is shown, although the text says: persists to 6 hrs, suggesting it is already there at 10 minutes, which is not the case.

We thank the referee for flagging this inconsistency. We have revised Figures 2D and 2E (now Figure 2B) to include multiple time points and extended incubation periods to match the kinetic studies done by ELISA (figure 1A). We added a 0 time point, incubation at 4°C (which was previously presented in supplementary Figure S3) that shows BDCA2 expression exclusively on the cell surface. We still have a 10 min incubation time point showing internalized BDCA2 that does not co-localize with LAMP1. We have now included a 1-hour incubation time point in which we observe clear co-localization of BDCA2 and LAMP-1. We confirmed persistent localization of BDCA2/24F4A complex in endolysosomal compartments at both 6 and 16 hour post-culture. The text has been revised to reflect those additions.

In Fig 2E the cells look really strange, please show cells with better morphology.

We have revised fig. 2E (now Figure 2B) to include cells with better morphology.

The text suggests it co-localizes with TLR9, please show this.

We have made extensive efforts to address this question. Latz et. al have previously shown that TLR9 is recruited from the ER to endolysosomal compartment after exposure of pDCs to CpG-A (Latz et al, 2004). Our data indicate that BDCA2 ligation leads to intracellular localization of BDCA2 and its retention in the endolysosomal compartment. These data raised the possibility that anti-BDCA2-mediated TLR9 inhibition might be facilitated by the internalization and localization of BDCA2 into endolysosomal compartments containing TLR9. To test this hypothesis confocal microscopy was used to follow the intracellular distribution of BDCA2 after 24F4A ligation. We followed the protocol described by Latz et. al. Cultured pDCs were incubated with AF647-labeled 24F4A, treated with the TLR9 ligand CpG-A and incubated at 37°C for 15 min. PDCs were then fixed and stained with fluorescently labeled antibodies to TLR9 (Figure for Referees R2). We observed a TLR9 staining pattern similar to that shown in Figure 2e of Latz et. al, shown below. Unfortunately, this TLR9 staining is very diffuse throughout the cytoplasm, which precluded us from confidently making a statement that TLR9 and BDCA2 were co-localized. Therefore, we have decided not to include these data. We are currently generating new anti-TLR9 mAbs which will hopefully be of better quality.

We have removed the comment about the likelihood of BDCA2 co-localization with TLR9 in the results section and now we only mention this hypothesis in the discussion section. Please note that for our confocal analysis we solicited the help of Prof. Tom Kirchhausen from Harvard Medical School, who is currently a visiting scientist at our Institution. Prof. Kirchhausen has an extensive experience in the use of confocal microscopy for solving complex biological questions. For his contribution Prof. Kirchhausen is appropriately mentioned in the acknowledgements.

Latz et. al 2004 Figure 2e

Figure R2. Purified human pDCs were cultured for 7 days as previously described by Latz et.al. (Latz et al, 2004). pDCs were incubated with AF647- labeled BIIB059 for 15 min at 37°C. During the last 10 min of the incubation, cells were treated with 1 μ M of the TLR9 ligand CpG-A. Cells were stained with fluorescently labeled antibodies to TLR9 (Anti-human TLR9 (eB72-1665, eBiosciences) and analyzed by confocal microscopy.

- Figure 3a is really hard to understand. Was the indirect and direct detections always combined?

We agree with the referee that Figure 3A as it was presented was difficult to understand. We have now modified it, as discussed below, to ensure a more clear representation of the data.

BDCA2 internalization in cynomolgus monkey whole blood is complicated by the fact that we don't have a non-cross blocking antibody that recognizes cynomolgus BDCA2. Therefore, we chose a two-step approach to detect internalization of BDCA2 from the surface of cynomolgus pDCs: Unoccupied surface BDCA2 was detected on pDCs using fluorescently-labeled 24F4A (direct method), while surface BDCA2 bound to 24F4A was detected using a fluorescently-labeled anti-human IgG1 (indirect method). The direct and indirect methods were run simultaneously, albeit in different staining cocktails, for every sample (pre- and post-dose) for all cynomolgus monkeys. We only inferred internalization when we didn't get a signal from the direct method (meaning lack of unoccupied BDCA2) coupled with loss of detectable 24F4A.

We revised the figure as follows:

- 1. Included separate panels for the direct and indirect methods.*
- 2. Distinguished the stain that depicts the maximal binding of 24F4A or "spiked 24F4A":
Maximal 24F4A binding was measured by "spiking-in" 10µg/ml of 24F4A in pre-dose cynomolgus whole blood in vitro and measuring BDCA2 bound by 24F4A using the indirect method (labeled anti-IgG1). This line is now represented by a solid red line.*
- 3. We included separate panels for pre-dose and post-dose samples.*

I understand there is internalization of BDCA2 after 6 h antibody treatment in vivo, but shouldn't the dotted line in fig3Aiii be negative, as in vehicle control with indirect label? Please clarify.

in the previous submission, the dotted line shown in figure 3Aiii (now represented in solid red line) corresponds to baseline 24F4A binding measurement determined by spiking 10µg/ml of 24F4A in pre-dose cynomolgus whole blood in vitro and measuring BDCA2 bound by 24F4A using the indirect method (labeled anti-IgG1). We agree with the referee that representing the measurement of baseline BDCA2 and maximal 24F4A binding using the same line color is confusing and has been rectified in the revised figure.

- there is no figure 3B-v, but it is referred to in the text:

We thank the referee for noticing this inconsistency, it should be 3B-iv, we have rectified it in the revised manuscript.

- please explain a bit more on MxA bioassay.

Additional details have been added to the materials and methods section in the revised manuscript.

- typo in sentence: MxA bioassay(..) that indirectly measure... should be measures:

This typo has been corrected.

Figure 3C: the decrease of IFN levels in the monkeys is not really impressive. What happened to Cyno A07858? Measurement stopped on day 14, but in figure 3Bi there are still data after this day.

We address this point in the supplementary Material and Methods " To compare type I IFN concentrations from IV 24F4A treated cohort, we used a two-way mixed effects analysis of variance (ANOVA) to fit to log10 values of type I IFN concentrations from samples obtained both prior to and including 31 days after intravenous dosing or the last day prior to loss of efficacy". For cynomolgus monkey A07858 the circulating 24F4A levels dropped after day 14. Concomitant with clearance of 24F4A there was a loss of the pharmacodynamics effect (BDCA2 internalization) indicating loss of efficacy.

We have now added a reference to that in the results section. In addition we moved the section "Statistical Analyses: Ex vivo TLR9 (CpG-A)-Induced type I IFN Production in Whole Blood Assay from Cynomolgus Monkeys Treated with 24F4A" from the supplementary Materials and Methods to the main text.

- It is interesting to see the effect on FcγRIIa. The authors claim this is the only Fc receptor of pDC, but they also express IIb (see review nature immunology 2014). It is hard to discriminate between IIa and b.

We thank the referee for raising this question. While CD32a was reported to be the only Fc receptor expressed on pDCs (Bave et al, 2003), recent transcript profiling data indicated that CD32b could be expressed at similar levels in pDCs (Guilliams, et al 2014) as the referee suggested. It is true that CD32a and b have high degree of homology therefore care should be taken when designing primers to detect CD32a vs. CD32b. To address this question, we evaluated the transcript levels of CD32a and CD32b in isolated pDCs using Q-PCR and confirmed the exclusive expression of CD32a on pDCs (Figure S8). In addition, functional data confirm the lack of expression of CD32b. CD32b engagement on B cells has been shown to inhibit signaling downstream from the B cell receptor. Therefore CD32b engagement is expected to affect the BDCA2-mediated BCR-like signaling cascade. CD32b engagement and impact on BDCA2 signaling would not have discriminated between immune complex-stimulation and stimulation with synthetic ligands. The fact that the added potency of 24F4A effector competent mAb is only seen with immune complex stimulation confirms that the effect is strictly mediated by the depletion of CD32a from the surface of pDCs.

- Fig 4C: legend does not match with X-axis (CPG vs R848).

This inconsistency has now been rectified in the legend.

The authors show that only with immunocomplexes and the BDCA2 antibody the FcR disappears. But how were the IC prepared? Explain

The M&M is not clear on this, what is the ratio between antibody and antigen? Add details

A better description of Sm/RNP immune complex preparation has now been added to the Materials and Methods section

- it is good that they do not only use an isotype control but also an CD40 antibody that can bind both with Fab and Fc part. This control antibody does not trigger internalization of FcR. However, this could be due to the expression level of the target, are BDCA2 and CD40 equally expressed? QIFI analysis of pDC would show this.

We agree with the referee that it is important to ascertain that the lack of an effect with anti-CD40 mAb is not due to lower CD40/CD32a ratio compared to BDCA2. We performed QIFI experiments that are now presented in supplementary Figure S10. The data show that both CD40 and BDCA2 are expressed at a higher level than CD32a (9 fold and 12 folds respectively). These expression levels would allow complete downmodulation of CD32a with either BDCA2 or CD40.

In light of the fact that BDCA2 density is 12-fold higher than CD32a, it is intriguing that CD32a and BDCA2 are downmodulated with same EC50 of internalization (Figure 5F). Therefore, for approximately every 12 BDCA2 molecules that get internalized, 1 CD32a gets internalized. This is presumably due to the relatively rapid internalization of BDCA2 (Figure. 2A) and the relatively low affinity of hIgG1 Fc for CD32a. The “scorpion mechanism” increases the avidity of 24F4A Fc for CD32a and therefore likelihood of engagement, but only if the molecules find each other within short amount of time after BDCA2 binding and before BDCA2 internalization.

- again, include the other BDCA2 antibodies for analyses of FcR internalization.

Data obtained with other BDCA2 antibodies have now been included in Figure 5 and Figure S9.

In the discussion, the authors mention that it was impossible to show co-localisation of CD32a and BDCA2. However, confocal picture of CD32a has been shown more often, so this should be possible and added to the paper.

We have put a lot of effort into optimizing the confocal experiment to evaluate the localization of CD32a and BDCA2. In these experiments, pDCs were added to fibronectin coated plates

and incubated with labeled 24F4A for 15 min at 37°C to allow internalization of the 2 receptors. The cells were then fixed, permeabilized and stained with fluorescently labeled anti-CD32a antibody (cat. # SAB4500869) that recognizes the C terminus of CD32a (Figure R3-A) or AT10 mAb that recognizes the N terminus of CD32a (Figure R3-B). Both antibodies were carefully titrated. As it can be seen in Figure R3, the intracellular staining of CD32a is very diffuse and possibly non-specific. Such diffuse CD32a staining makes it impossible to conclude with certainty that BDCA2 and CD32a are co-localized.

Also, does the same mechanism of CD32a internalization happens in cynomolgus monkeys?

To address the referee's question we performed experiments to determine whether 24F4A can lead to CD32a internalization on cynomolgus monkey pDCs. We previously determined that the binding of human IgG1 Fc was equivalent to both human and cynomolgus CD32a (our unpublished observations). We have observed that cynomolgus monkeys have on average 2-fold less circulating pDCs compared to humans (our unpublished observations). The lower numbers of pDCs combined with the limited blood volume we could get from cynomolgus monkeys precluded us from isolating cynomolgus pDCs. To this end, we determined BDCA2 and CD32a levels in the presence of 24F4A vs. 24F4A-ef in human and cynomolgus monkey whole blood assays. In this experiment, human or cynomolgus whole blood was treated with increasing concentration of 24F4A or 24F4-ef and incubated for 16 hours at 37° C. Surface BDCA2 and CD32a were assessed using flow cytometry. In

human whole blood assays, we successfully reproduced the data seen in isolated pDCs; CD32a was downmodulated upon treatment with 24F4A but not with 24F4A-ef.

In cynomolgus monkey we observed a sharp decrease in pDC viability after incubating whole blood for 16 h at 37° C that was not observed in human whole blood assays. While shorter incubation periods preserved the viability of cynomolgus pDCs it did not lead to appreciable 24F4A-mediated BDCA2 internalization that could support CD32a downmodulation as we have seen in the human in vitro system. Due to these technical considerations we don't believe we have a good in vitro system to address whether the CD32a downmodulation mechanism is operative in cynomolgus monkeys.

Please include reference for 'scorpion' mechanism with first sentence. Isn't this the same as described by Kurlander a long time ago?

We agree that this reference should have been included when we first referenced the scorpion mechanism. We added this reference.

References: I think the nice overview of IFN directed treatment of SLE by Kirou in clinical Immunology of 2013 should be included.

This reference has been added in the discussion section, we thank the referee for suggesting it. We have now added a paragraph in the discussion section in which we compare the therapeutic approach of targeting pDCs with the anti-IFN strategies and we cite this review when we refer to the different approaches used to date for targeting the IFN-I pathway in SLE.

References

Bave U, Magnusson M, Eloranta ML, Perers A, Alm GV, Ronnblom L (2003) Fc gamma RIIa is expressed on natural IFN-alpha-producing cells (plasmacytoid dendritic cells) and is required for the IFN-alpha production induced by apoptotic cells combined with lupus IgG. *Journal of immunology* **171**: 3296-3302

Hare JF, Taylor K (1991) Mechanisms of plasma membrane protein degradation: recycling proteins are degraded more rapidly than those confined to the cell surface. *Proceedings of the National Academy of Sciences of the United States of America* **88**: 5902-5906

Latz E, Schoenemeyer A, Visintin A, Fitzgerald KA, Monks BG, Knetter CF, Lien E, Nilsen NJ, Espevik T, Golenbock DT (2004) TLR9 signals after translocating from the ER to CpG DNA in the lysosome. *Nature immunology* **5**: 190-198

2nd Editorial Decision

22 January 2015

Thank you for the submission of your revised manuscript to EMBO Molecular Medicine. We have now received the enclosed reports from the referees who were asked to re-assess it. As you will see, the reviewers are now globally supportive and I am pleased to inform you that we will be able to accept your manuscript pending the following final amendment:

Please address the last minor comment from referee 1.

Please submit your revised manuscript within two weeks. I look forward to seeing a revised form of your manuscript as soon as possible.

***** Reviewer's comments *****

Referee #1 (Comments on Novelty/Model System):

I think most of the results are still confirmatory in nature but the authors have made clearer what is new and what has been shown before:

Figure 1: As the authors admit, essentially confirmatory to previous findings. New aspects are the utilization of whole blood and the correlation analysis of the IC50 of IFN-alpha inhibition and the EC50 of BDCA-2 internalization.

Figure 2: As the authors admit, essentially confirmatory to previous findings. New aspects are the lack of reappearance of BDCA-2 on the cell surface after internalization within 16 hours and the inhibitory signaling by pre-incubation with anti BDCA-2 mAb before stimulation of pDCs.

Figure 3: first study in cynomolgus monkeys, but another in vivo model has been published before.

Figures 4 and 5: Basically the only really new finding: wildtype IgG1 anti BDCA-2 is more potent than Fc-effectorless anti BDCA-2 at inhibiting type I interferon induction by immune complexes. This is most likely due to additional down-modulation of CD32a on PDCs by anti BDCA-2 mAb-induced co-internalization of BDCA-2 and CD32a.

Referee #1 (Remarks):

The authors have adequately addressed my concerns on the novelty of their findings and put more emphasis on previous findings.

Minor mistake:

Table 1 shows an EC50 for AC144-binding to cynomolgus BDCA-2 of 11,047 mg/ml! However, according to Figure S1 the EC50 for AC144-binding to cynomolgus BDCA-2 is rather in the range of 5 microgram/ml (2000 times lower).

Referee #2 (Remarks):

The manuscript has been adequately revised and appears suitable for publication

Referee #3 (Remarks):

The Manuscript has clearly improved by the revision. I have no further concerns.